# TEST-TIME ADAPTATION AGAINST MULTI-MODAL RELIABILITY BIAS

**Mouxing Yang**[1]     **Yunfan Li**[1]     **Changqing Zhang**[2]     **Peng Hu**[1]     **Xi Peng**[1*]

Sichuan University[1]     Tianjin University[2]

{yangmouxing, yunfanli.gm, penghu.ml, pengx.gm}@gmail.com,
zhangchangqing@tju.edu.cn

## ABSTRACT

Test-time adaptation (TTA) has emerged as a new paradigm for reconciling distribution shifts across domains without accessing source data. However, existing TTA methods mainly concentrate on uni-modal tasks, overlooking the complexity of multi-modal scenarios. In this paper, we delve into the multi-modal test-time adaptation and reveal a new challenge named reliability bias. Different from the definition of traditional distribution shifts, reliability bias refers to the information discrepancies across different modalities derived from intra-modal distribution shifts. To solve the challenge, we propose a novel method, dubbed REliable fusion and robust ADaptation (READ). On the one hand, unlike the existing TTA paradigm that mainly repurposes the normalization layers, READ employs a new paradigm that modulates the attention between modalities in a self-adaptive way, supporting reliable fusion against reliability bias. On the other hand, READ adopts a novel objective function for robust multi-modal adaptation, where the contributions of confident predictions could be amplified and the negative impacts of noisy predictions could be mitigated. Moreover, we introduce two new benchmarks to facilitate comprehensive evaluations of multi-modal TTA under reliability bias. Extensive experiments on the benchmarks verify the effectiveness of our method against multi-modal reliability bias. The code and benchmarks are available at https://github.com/XLearning-SCU/2024-ICLR-READ.

## 1 INTRODUCTION

Multi-modal pre-trained models (Radford et al., 2021; Girdhar et al., 2023; Li et al., 2023; Lin et al., 2024) have shown great potential in various applications, being research focuses in both academic and industrial communities. After acquiring common knowledge from the source domain, pre-trained models could be customized into specific tasks through the attention mechanism (Vaswani et al., 2017; Gong et al., 2023) that integrates knowledge from different modalities in the target domain. Although such a paradigm has achieved promising performance, its success heavily relies on the identical distribution between the source domain and target/test domain (Chen et al., 2023). However, as shown in Fig. 1(a), it is daunting to meet such a mild assumption, especially in open-world scenarios with unpredictable factors such as the changing weather (*e.g. fog*) and degenerated sensors (*e.g. defocus*) would lead to the distribution shifts (Hendrycks & Dietterich, 2019).

Toward achieving robustness against distribution shifts, numerous test-time adaptation (TTA) methods have been proposed (Wang et al., 2021; Yu et al., 2023a; Niu et al., 2022). Most of them work by updating parameters of normalization layers in the source model (Ioffe & Szegedy, 2015; Ba et al., 2016), hoping to bridge the gaps between domains (Schneider et al., 2020; Zhang et al., 2022; Nado et al., 2020; Hu et al., 2021; Iwasawa & Matsuo, 2021). To this end, they usually minimize the entropy-based objective on the model predictions of unlabeled test samples. Despite the significant success, almost all existing TTA methods are devoted to handling distribution shifts between domains while ignoring specific challenges in multi-modal learning scenarios. Specifically, once some modalities are contaminated with distribution shifts, the information discrepancies between modalities would be enlarged, leading to reliability bias across modalities. For example, as shown

---

*Corresponding author.

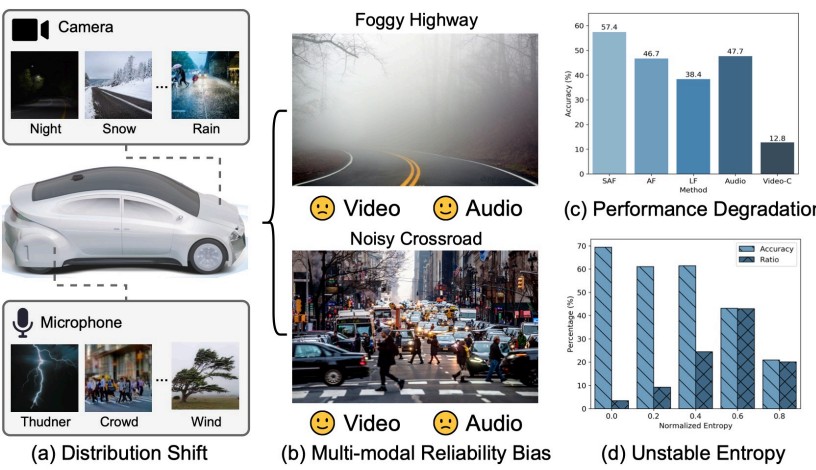

Figure 1: Our observations. (a) **Distribution shift**: some sensors of autonomous vehicles might encounter different situations in the wild, leading to domain shifts in certain modalities. (b) **Multi-modal reliability bias**: due to the distribution shifts, some corrupted modalities will lose the task-specific information and suffer from reliability bias during cross-modal fusion compared to the un-corrupted counterparts. (c) **Performance degradation**: the video modality contaminated with relia-bility bias (Video-C) has poor recognition accuracy compared to the audio modality. Both vanilla attention-based fusion (AF) and late fusion (LF) manner give inaccurate predictions compared to the single-modality ones. Instead, the proposed self-adaptive attention-based fusion (SAF) could achieve reliable fusion thus guaranteeing the performance gain in multi-modal scenarios. (d) **Unsta-ble entropy**: once the more informative modalities are corrupted (*e.g.* video for action recognition), it would be challenged to give accurate predictions. Consequently, the entropy of multi-modal pre-dictions would be unstable. In other words, the ratio of confident predictions would decrease while the noise might dominate the predictions. All results in the figures are from experiments conducted on a subset of the Kinetics dataset (Kay et al., 2017) with foggy corruption in the video modality.

in Fig. 1(b), when an autonomous vehicle equipped with camera and audio sensors drives into a foggy highway or noisy crossroad, either the visual or audio modalities would be corrupted. As a result, the reliability balance across the modalities will be destroyed, and the performance of the model would heavily degrade if each modality is equally treated as depicted in Fig. 1(c). Although some studies (Zhang et al., 2023; Peng et al., 2022) have been conducted toward imbalanced multi-modal learning, they mainly focus on altering the training process with the labeled samples in the source domain, rather than adapting biased modalities during the test time.

Based on the above observations, in this paper, we reveal a new problem for multi-modal test-time adaptation, *i.e.*, reliability bias. Different from the distribution shifts between domains, reliability bias refers to the information discrepancies across different modalities derived from the intra-modal distribution shifts. It should be pointed out that it is intractable to conquer the reliability bias problem using the existing TTA methods (Niu et al., 2023; Shin et al., 2022) due to the following reasons. First, it is impossible to completely reconcile the distribution shifts through updating the parameters of normalization layers. As a result, it is inevitable to introduce reliability bias across modalities. Second, as shown in Fig. 1(d), in the multi-modal scenarios, once the representative modality is corrupted, noisy predictions would dominate the adaptation process. As a result, simply minimizing the entropy on all predictions or only confident predictions might either lead to model overfitting or underfitting on the test data. To support our claims, we provide some empirical results in Section 4.2 and Fig. 3.

To achieve reliability-bias robust multi-modal TTA, we propose a novel method, dubbed REliable fusion And robust ADaptation (READ). READ handles the reliability bias challenge by resorting to the following two-fold modules. On the one hand, instead of reconciling the intra-modality distri-bution shifts through repurposing normalization layers, we propose modulating the attention-based fusion layers in a self-adaptive manner for reliable cross-modal fusion during test time. On the other hand, we design a novel objective function for robust multi-modal adaptation. In short, the objec-

tive function can not only amplify the contributions of confident predictions but also prevent noisy predictions from dominating the adaptation process.

The major contributions and novelties of this work could be summarized as follows:

1. We reveal a new challenge for multi-modal test-time adaptation, *i.e.*, reliability bias. In a word, reliability bias refers to the information discrepancies across different modalities, derived from the distribution shifts between domains.

2. To enjoy robustness against reliability bias, we propose a novel method named READ. Unlike most existing TTA methods that reconcile the distribution shifts by repurposing the normalization layers, READ achieves reliable fusion and robust adaptation by modulating the attention-based fusion layers in a self-adaptive manner under the support of a novel objective function.

3. We provide two benchmarks (multi-modal action recognition and event classification) for multi-modal TTA with reliability bias. Extensive experiments on the benchmarks not only verify the effectiveness of our method but also give some observations for the community.

## 2 RELATED WORK

In this section, we briefly review some related topics to this work, *i.e.*, test-time adaptation, and imbalanced multi-modal learning.

### 2.1 TEST-TIME ADAPTATION

Test-time adaptation aims at bridging the gaps between source and target domains during test time without accessing the source data. Toward this goal, some test-time training methods (Liu et al., 2021; Sun et al., 2020) have been proposed, which additionally add a self-supervised task in the training process. As a result, the source model could be adapted by performing the self-supervised task on test samples. Such a paradigm needs to alter the training process and might be limited in the pre-trained model era. To remedy this, the fully test-time adaptation paradigm has emerged and plenty of methods have been proposed in recent years, which could be roughly divided into the following categories. i) online TTA methods (Wang et al., 2021; Gao et al., 2023), which updates the specific model parameters (always the normalization layers) with the coming test samples by resorting to some unsupervised objectives such as entropy minimization on predictions. ii) robust TTA methods (Niu et al., 2023; Zhou et al., 2023), which considers some challenging and practical adaptation settings such as label shifts, single sample, mixed domain shifts, etc. iii) Continual TTA methods (Gan et al., 2023; Wang et al., 2022) which aims to solve the continual and changing shifts along test time. iv) TTA beyond recognition (Shin et al., 2022; Lee et al., 2023), which focuses on applications beyond image classification such as multi-modal segmentation, pose estimation.

In this paper, we focus on online TTA and aim to achieve multi-modal test-time adaptation against modality reliability bias. Among the existing TTA studies, MM-TTA (Shin et al., 2022) might be most relevant to our work, while having the following main differences. i) Problem/motivation differences. MM-TTA focuses on reconciling the distribution shifts between domains for 2D-3D joint segmentation tasks. In contrast, this work aims to handle the modality reliability bias challenge overlooked by the existing studies and validate the necessity and effectiveness in multi-modal scenarios including audio-video event classification, and action recognition. ii) Approach/paradigm differences. MM-TTA achieves adaptation by updating the normalization layer like most existing TTA methods, while this work proposes modulating the attention-based fusion layers in a self-adaptive way. iii) Objective function differences. MM-TTA adopts a noise-filter cross-entropy loss whose pseudo labels are selected based on a set of slow-fast models. In contrast, we design a novel confidence-aware objective function, which would not only benefit the model optimization by exploiting the confident predictions but also hinder the model from overfitting noise.

### 2.2 IMBALANCED MULTI-MODAL LEARNING

Multi-modal learning has emerged as a promising avenue for understanding the world, encompassing various tasks such as recognition, clustering, and retrieval across diverse views, media, or domains (Lin et al., 2021; Yang et al., 2022; 2021). Some recent studies have found that multi-modal

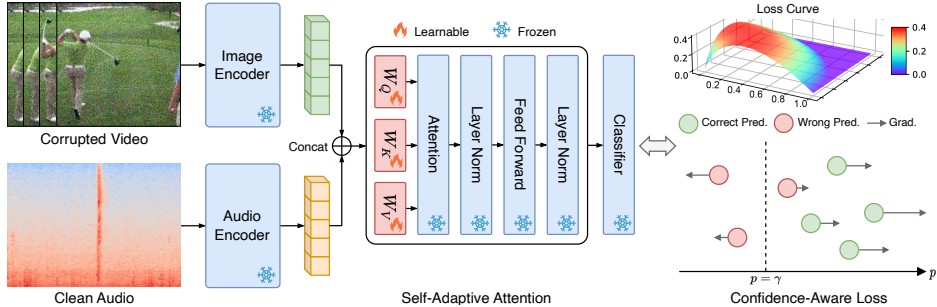

Figure 2: The pipeline of our method (READ). On the adaptation process, biased modality (*e.g.* corrupted video) and unbiased modality (*e.g.* clean audio) are input into two modality-specific encoders, and the output embeddings are concatenated at token-level for fusion. During the cross-modal fusion, the attention is calculated between the token embeddings in a self-adaptive manner and severed as the reliability for fusion. After that, the multi-modal predictions are obtained by the classifier with the fused embeddings. The confidence-aware loss will amplify the contributions of the high-confident predictions ($p > \gamma$) by increasing the gradients and alleviate the influence of low-confident predictions ($p < \gamma$) by reversing the gradients. Both the amplification and alleviation effects are proportional to the confidences of predictions.

learning might not achieve better performance compared to the single-modal counterparts (Peng et al., 2022; Wang et al., 2020; Fan et al., 2023; Du et al., 2021; Wei et al., 2022). The essence behind the problem could be boiled down to the discrepancy/imbalance between modalities. Wang et al. (2020) observes that different modalities have various convergence rates. Motivated by the observation, Peng et al. (2022) makes deep analysis and finds that some modalities embrace more task-specific information under certain scenarios, *e.g.*, audio for multi-modal sound localization. As a result, the more informative modalities might dominate the learning process, thus hindering the fitting of other modalities if each modality is equally during the optimization. Toward guaranteeing the performance gain of multi-modality learning, some works have delved into learning with imbalanced multi-modal data. Wang et al. (2020) uses auxiliary networks to penalize the modalities by considering their overfitting behaviors, leading to a better fusion between modalities. Peng et al. (2022) designs a gradient-modulation strategy that adaptively adjusts the gradients for different modalities according to their contributions to the network optimization. From the perspective of uncertainty learning, Zhang et al. (2023) proposes a robust multimodal fusion method with theoretical guarantees, which estimates the certainty of each modality and accordingly achieves weighty cross-modal fusion.

In this paper, we focus on the test-time reliable fusion across modalities, and the setting is significantly different from the existing imbalanced multi-modal learning studies. Specifically, the existing studies focus on learning with unbalanced modalities under the labeled source domain, while this work aims to adapt the source model with unlabeled multi-modal test pairs during test time.

## 3 METHOD: RELIABLE FUSION AND ROBUST ADAPTATION

In this section, we elaborate on the proposed method dubbed reliable fusion and robust adaptation (READ) for multi-modal test-time adaptation against reliability bias. As shown in Fig. 2, READ consists of the self-adaptive attention module to achieve reliable fusion across different modalities, and the confidence-aware loss function for robust test-time adaptation. In the following, we first present the used notations and definition of the multi-modal reliability bias problem in Section 3.1, then introduce the self-adaptive attention module in Section 3.2, and finally elaborate on the confidence-aware loss function in Section 3.3.

### 3.1 NOTATIONS AND PROBLEM FORMULATION

Without loss of generality, we take two modalities as a showcase for clarity of presentation. For clarity, we use $F_{\Theta_s} = \{f_{\Theta_s^a}, f_{\Theta_s^v}, f_{\Theta_s^m}, C_{\Theta_s}\}$ to denote the source model that trained on the labeled

training set $\{\mathbf{s}_i^a, \mathbf{s}_i^v, y_i\}_{i=1}^{N_s}$, where $f_{\Theta_s^a}$ and $f_{\Theta_s^v}$ are the specific Transformer encoders for modality $a$ and $v$, $f_{\Theta_s^m}$ and $C_{\Theta_s}$ are the multi-modal fusion layer and the following classifier, and $\mathbf{s}_i^h$ consists of tokens $\{\mathbf{s}_{ij}^h\}_{j=1}^{T^h}$ ($h \in \{a, v\}$). During training, the source model $F_{\Theta_s}$ would (over)fit the distribution of training data, $i.e.$, $P(\mathbf{s})$. As a result, in the inference/testing stage, the performance of $F_{\Theta_s}$ would heavily degrade once the distribution shifts emerge due to weather changes, sensor degeneration, etc. In other words, $P(\mathbf{s}) \neq P(\mathbf{x})$ where $\mathbf{x}$ are the unlabeled test multi-modal data. Test-time adaptation (TTA) (Yu et al., 2023a; Wang et al., 2021) aims to quickly reconcile the shifts for coming test samples through online updating the parameter of $F$ from $\Theta_s$ to $\Theta$ during test-time. To this end, most of them minimize the objective function below:

$$\min_{\widetilde{\Theta}} \mathcal{L}^{tta}(\mathbf{p}),  \qquad (1)$$

where $\mathcal{L}^{tta}$ is the loss function, $\tilde{\Theta} \subseteq \Theta$ denotes the learnable parameters (usually BN or LN) of the adapted model $F_{\Theta}$, $\mathbf{p}$ is the predictions of $F_{\Theta}$ for test-time multi-modal pairs $(\mathbf{x}^a, \mathbf{x}^v)$, $i.e.$, $\mathbf{p} = C_{\Theta_s}(f_{\Theta^m}(\mathbf{z}^a, \mathbf{z}^v))$ where $\mathbf{z}^a = f_{\Theta^a}(\mathbf{x}^a)$ and $\mathbf{z}^v = f_{\Theta^v}(\mathbf{x}^v)$. Under the unlabeled setting, most TTA methods usually use the entropy minimization objective as $\mathcal{L}^{tta}$ or design another variants.

Although existing TTA methods have achieved great success, most of them focus on single-modality tasks and cannot handle the reliability bias problem in multi-modal scenarios. Concretely, in the wild, the corrupted modalities will lose some task-specific information compared to the other clean ones as discussed in Introduction. As a result, the clean modalities will be more reliable than the corrupted ones during the cross-modal fusion, $i.e.$, the modality reliability bias. In Experiment, we empirically validate that the reliability bias problem would significantly degrade the performance of existing TTA methods. Therefore, our goal becomes achieving cross-modal fusion on the reliability biased modalities and performing robust test-time adaptation.

## 3.2 RELIABLE FUSION

To fuse the information across modalities, one widely-used solution is the late-fusion-based manner (Peng et al., 2022; Zhang et al., 2023). Mathematically, given the test-time embeddings $(\mathbf{z}^a, \mathbf{z}^v)$, the multi-modal predictions $\mathbf{p}^{\mathrm{lf}}$ obtained via late fusion could be formulated as follows,

$$\mathbf{p}^{\mathrm{lf}} = (C_{\Theta}^a(\mathrm{mean}(\mathbf{z}^a)) + C_{\Theta}^v(\mathrm{mean}(\mathbf{z}^v)))/2,  \qquad (2)$$

where $\mathrm{mean}$ denotes the token-wise mean operation, and $C_{\Theta}^a$ and $C_{\Theta}^v$ are two modality-specific classifiers. Clearly, the late fusion manner equally treats each modality whether they are reliable or not, being sensitive to the reliability bias problem as depicted in Fig. 4.

As a remedy, we propose the self-adaptive attention module to dynamically integrate information from different modalities. Specifically, the modality-specific embeddings $\mathbf{z}^a$ and $\mathbf{z}^v$ are first concatenated at token-level, then projected into the query, key, and value matrixes. More formally,

$$\begin{aligned}
\mathbf{Q} &= W_{\Theta^Q}([\mathbf{z}^a; \mathbf{z}^v]) + B_{\Theta^Q}, \\
\mathbf{K} &= W_{\Theta^K}([\mathbf{z}^a; \mathbf{z}^v]) + B_{\Theta^K}, \\
\mathbf{V} &= W_{\Theta^V}([\mathbf{z}^a; \mathbf{z}^v]) + B_{\Theta^V},
\end{aligned}  \qquad (3)$$

where $W_{\Theta^h}$ and $B_{\Theta^h}$ ($h \in \{Q, K, V\}$) are the projector and bias term inherited from the source model and updated during the test time, and $[\cdot; \cdot]$ denote the token-level concatenate operation. After that, the attention map could be calculated as follows,

$$\mathbf{A} = \mathrm{Softmax}\left(\frac{\mathbf{Q}\mathbf{K}^T}{\sqrt{d}}\right),  \qquad (4)$$

where the cell of $\mathbf{A}_{rt}$ denote the similarity value between the $r$-th token and $t$-token from $\mathbf{z}^h$ ($h \in \{a, v\}$), and $d$ is the latent dimension of the tokens.

Note that, the vanilla attention mechanism (Vaswani et al., 2017; Gong et al., 2023) usually keeps the parameters inherited from the source model, and performs information integration across modalities. Apparently, the distribution shift between training and test-time data might hinder the similarity estimation between tokens. As a result, reliable fusion on biased modalities cannot be guaranteed. Instead, we hope that the model could focus more on the unbiased modalities and avoid the interoperation from the bias. To this end, we propose repurposing the cross-modal attention-based fusion

layers in a self-adaptive way. In other words, the parameters of $W_{\Theta^h}$ and $B_{\Theta^h}$ ($h \in \{Q, K, V\}$) would be updated in order to adapt the test-time distribution. Thanks to the attention modulation, the model would focus more on the unbiased modalities, leading to reliable cross-modal fusion during test time. We empirically verify the above claims in Fig. 3.

Finally, the predictions obtained through our self-adaptive attention-based fusion layer could be formulated as below,

$$\mathbf{p}^{\text{saf}} = C_{\Theta_s} \left( \text{mean} \left( \mathbf{AV} \right) \right). \tag{5}$$

## 3.3 ROBUST ADAPTATION

After cross-modal fusion on modalities with reliability bias, the challenge becomes achieving robust adaptation against the distribution shifts in multi-modal scenarios. One feasible solution is adopting the widely-used entropy minimization objective on either all predictions (Wang et al., 2021) or only some high-confident ones (Niu et al., 2023). However, as discussed in Introduction, once some informative modalities (*e.g.*, visual modality for action recognition task) are corrupted, the overall task-specific information would greatly reduce. As a result, the accuracy of the predictions from the source model might decrease. At this time, either the vanilla entropy minimization objective or the noise-filter one would overfit the noisy predictions or underfit clean predictions, leading to degraded adaptation effects as verified in Fig. 1(d) and Tables 1-3.

As a remedy, we propose a novel confidence-aware loss function for robust adaptation. Formally, given a mini-batch test predictions of size $B$, the loss function is designed as below:

$$\mathcal{L}_{ra} = \frac{1}{B} \sum_{i=1}^{B} p_i \log \left( \frac{e\gamma}{p_i} \right), \tag{6}$$

where $p_i$ is confidence of the prediction $\mathbf{p}_i^{\text{saf}}$, *i.e.*, $p_i = \max \left( \delta \left( \mathbf{p}_i^{\text{saf}} \right) \right)$, $\delta$ is the softmax operation and $\gamma$ is a threshold for confident prediction division fixed as a constant in all our experiments. In the following, we mathematically show that why the loss function could achieve robust adaptation. To begin with, we first plot the loss performance curve for clarity. As shown in Fig. 2, the loss embraces the following merits.

**Remark 1.** $\mathcal{L}_{ra}$ *will reduce non-monotonously for different predictions. Consequently, the high-confident predictions ($p_i > \gamma$, possible clean) will contribute to optimization while the influence of low-confident predictions ($p_i < \gamma$, possible noisy) will be eliminated. Meanwhile, contributions of the high-confident predictions will be amplified with the increasing confidences, while the negative impacts of the low-confidence ones will be reduced with the decreasing confidences.*

The above properties of our loss could be supported by the following Theorems.

**Theorem 1.** *The gradient direction produced by $\mathcal{L}_{ra}$ is non-monotonous.*

**Theorem 2.** *The gradient value will rise with increasing $p_i$ i.f.f. $p_i \in (\gamma, 1)$ or decreasing $p_i$ i.f.f. $p_i \in (0, \gamma)$.*

Due to the space limitation, we remove the according proofs into the Appendix. Thanks to the favorable properties of $\mathcal{L}_{ra}$, the robust adaptation could be achieved. On the one hand, the model would not overfit the low-confident predictions, thus preventing the noise from dominating the adaptation process. On the other hand, the model will focus more on the high-confident predictions, thus benefiting the optimization.

Combining both the self-adaptive attention module and the confidence-aware loss function together, we could obtain the final objective function for multi-modal test-time adaptation as follows,

$$\min_{\widehat{\Theta}} \mathcal{L}(\mathbf{z}_{\text{af}}), \tag{7}$$

where $\widehat{\Theta} = \{\Theta^Q, \Theta^K, \Theta^V\} \subseteq \Theta$, $\mathcal{L} = \mathcal{L}_{ra} + \mathcal{L}_{bal}$, and $\mathcal{L}_{bal} = \sum_{k=1}^{K} \delta \left( c^k \right) \log \delta \left( c^k \right)$ is an negative entropy loss term to make the prediction balance following Yu et al. (2023b); Zhou et al. (2023) where $c^k = \sum_{i=1}^{B} \delta \left( \mathbf{p}_i^{\text{saf}} \right)$ and $K$ is the class number.

Table 1: Comparisons with SOTA methods on Kinetics50-C benchmark with **corrupted video modality (severity level 5)** regarding the accuracy (%) metric. "Stat." and "Dyn." are the abbreviation of "Statical" and "Dynamic", while "LN", "LF", "AF" and "SAF" denotes the layer normalization, late fusion, attention-based fusion, and self-adaptive attention-based fusion, respectively. The results are the mean values among 5 random seeds, and the best results are highlighted in **bold**.

| Methods | Noise | | | Blur | | | | Weather | | | | Digital | | | | Avg. |
|---|---|---|---|---|---|---|---|---|---|---|---|---|---|---|---|---|
| | Gauss. | Shot | Impul. | Defoc. | Glass | Mot. | Zoom | Snow | Frost | Fog | Brit. | Contr. | Elas. | Pix. | JPEG | |
| Source ((Stat. LN) & LF) | 31.8 | 33.4 | 31.7 | 64.0 | 54.3 | 67.5 | 61.9 | 50.9 | 54.8 | 38.4 | 72.3 | 44.0 | 60.2 | 61.7 | 56.4 | 52.2 |
| • MM-TTA (Dyn. LN) | 46.2 | 46.6 | 46.1 | 58.8 | 55.7 | 62.6 | 58.7 | 52.6 | 54.4 | 48.5 | 69.1 | 49.3 | 57.6 | 56.4 | 54.6 | 54.5 |
| • Tent (Dyn. LN) | 28.6 | 29.8 | 28.3 | 63.4 | 51.1 | 67.7 | 61.7 | 46.5 | 51.3 | 24.5 | 72.3 | 38.6 | 60.7 | 61.8 | 54.9 | 49.4 |
| • EATA (Dyn. LN) | 31.8 | 33.3 | 31.6 | 64.2 | 54.6 | 67.7 | 62.2 | 51.3 | 54.7 | 38.1 | 72.5 | 44.2 | 60.4 | 62.0 | 57.0 | 52.4 |
| • SAR (Dyn. LN) | 31.9 | 33.3 | 31.7 | 63.8 | 54.0 | 67.7 | 61.8 | 50.7 | 54.5 | 38.8 | 72.3 | 44.0 | 60.3 | 62.0 | 56.5 | 52.2 |
| • READ (Dyn. LN) | 34.0 | 34.5 | 33.8 | 65.3 | 57.7 | 68.7 | 64.9 | 56.1 | 57.5 | 41.1 | 73.2 | 48.7 | 62.9 | 64.6 | 59.2 | 54.8 |
| Source (Stat. (LN&AF)) | 46.8 | 48.0 | 46.9 | 67.5 | 62.2 | 70.8 | 66.7 | 61.6 | 60.3 | 46.7 | 75.2 | 52.1 | 65.7 | 66.5 | 61.9 | 59.9 |
| • Tent (Dyn. LN) | 46.3 | 47.0 | 46.3 | 67.2 | 62.5 | 71.0 | 67.6 | 63.1 | 61.1 | 34.9 | 75.4 | 51.6 | 66.8 | 67.2 | 62.7 | 59.4 |
| • EATA (Dyn. LN) | 46.8 | 47.6 | 47.1 | 67.2 | 62.7 | 70.6 | 67.2 | 62.3 | 60.9 | 46.7 | 75.2 | 52.4 | 65.9 | 66.8 | 62.5 | 60.1 |
| • SAR (Dyn. LN) | 46.7 | 47.4 | 46.8 | 67.0 | 61.9 | 70.4 | 66.4 | 61.8 | 60.6 | 46.0 | 75.2 | 52.1 | 65.7 | 66.4 | 62.0 | 59.8 |
| • READ (SAF) | **49.4** | **49.7** | **49.0** | **68.0** | **65.1** | **71.2** | **69.0** | **64.5** | **64.4** | **57.4** | **75.5** | **53.6** | **68.3** | **68.0** | **65.1** | **62.5** |

## 4 EXPERIMENTS

In this section, we evaluate the proposed READ on the audio-visual joint action recognition and event classification tasks under multi-modal TTA with reliability bias. The organization of this section is as follows. In Section 4.1, we present the experiment settings including benchmark construction and implementation details. In Section 4.2, we compare READ with the state-of-the-art (SOTA) TTA methods under different settings, revealing some observations. In Section 4.3, we perform ablation studies and analytic experiments to give a comprehensive understanding on READ.

### 4.1 EXPERIMENT SETTINGS

To facilitate the investigation of multi-modal TTA with reliability bias, we construct two benchmarks based on the widely-used multi-modal datasets Kinetics (Kay et al., 2017) and VGGSound (Chen et al., 2020). For comprehensive studies, following Hendrycks & Dietterich (2019), we introduce 15 types of corruptions for the video modality and 6 for the audio modality. Each type of corruption has five levels of severity. As a result, we obtain Kinetics50-C and VGGSound-C benchmarks with either corrupted audio or corrupted video modalities. READ is a general framework that could endow most existing visual-audio pre-trained models with robustness against reliability bias. Without loss of generality, we choose the SOTA CAV-MAE (Gong et al., 2023) model pre-trained on web-scale audio-visual data as the backbone and fine-tune it on the training sets of Kinetics50 and VGGSound dataset, obtaining the corresponding source models. In other words, the training sets of Kinetics50 and VGGSound are the source domains while Kinetics50-C and VGGSound-C are the target domains. During the test-time adaptation phase, READ conducts online updates on specific parameters of the source models using the Adam optimizer. This process utilizes an initial learning rate of 0.0001 for every mini-batch of size 64 within a single epoch. The confidence threshold $\gamma$ in Eq. 6 is fixed as $e^{-1}$ for all settings. All evaluations are run on Ubuntu 20.04 platform with NVIDIA 3090 GPUs. Due to the space limitation, we remove more details into Appendixes B and C.

### 4.2 COMPARISONS WITH STATE-OF-THE-ARTS

We compare READ with four SOTA TTA methods including Tent (Wang et al., 2021), MMT (Shin et al., 2022), EATA (Niu et al., 2022), and SAR (Niu et al., 2023) under different settings. The results are presented in Tables 1-3 and 10-12 wherein the two blocks denote the source model trained with late fusion (Eq. 2) and attention-based fusion, respectively. From the results, one could have the following observations and conclusions.

- TTA methods using late fusion are most sensitive to the reliability bias, which could be attributed to the equal treatments on each modality. MM-TTA with a carefully-designed pseudo label generation strategy for late fusion cannot always achieve robustness, especially when the representative modality is biased (*e.g.*, audio for VGGSound).

- The attention-based fusion can improve the robustness against reliability bias compared to late fusion. However, TTA methods using vanilla attention-based fusion can only achieve

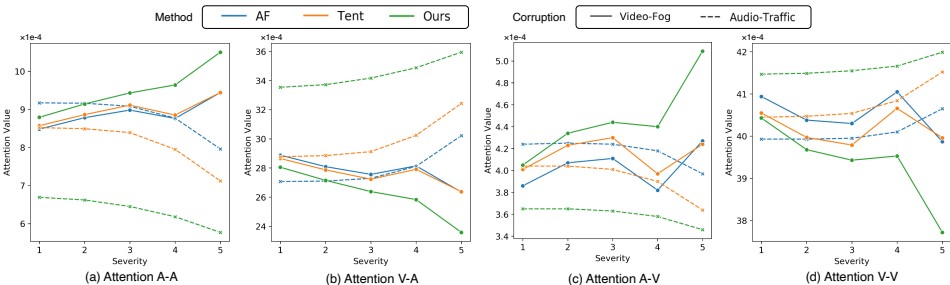

Figure 3: Reliable fusion with respect to attention values. In the figure, "Tent" and "AF" denote the variants of adopting vanilla attention-based fusion with and without parameter updating of LN, respectively. "Attention X-Y" (X, Y $\in$ {A, V}) indicates the configuration where query and key correspond to the tokens of modality X and modality Y in Eq. 4, respectively.

Table 2: Comparisons with SOTA methods on Kinetics50-C (left part) and VGGSound-C (right part) benchmarks with **corrupted audio modality (severity level 5)**.

| Methods | Noise | | | Weather | | | | Noise | | | Weather | | | |
|---|---|---|---|---|---|---|---|---|---|---|---|---|---|---|
| | Gauss. | Traff. | Crowd. | Rain | Thund. | Wind | Avg. | Gauss. | Traff. | Crowd. | Rain | Thund. | Wind | Avg. |
| Source ((Stat. LN) & LF) | 71.1 | 67.8 | 67.4 | 67.4 | 70.6 | 68.6 | 68.8 | 29.5 | 17.1 | 22.6 | 17.3 | 33.7 | 20.6 | 23.5 |
| • MM-TTA (Dyn. LN) | 70.8 | 69.2 | 68.5 | 69.0 | 69.8 | 69.4 | 69.4 | 14.1 | 5.2 | 6.4 | 6.9 | 8.6 | 4.5 | 7.6 |
| • Tent (Dyn. LN) | 71.1 | 68.6 | 67.8 | 67.4 | 71.2 | 68.9 | 69.2 | 6.4 | 2.1 | 2.9 | 1.9 | 9.5 | 3.1 | 4.3 |
| • EATA (Dyn. LN) | 71.2 | 67.9 | 67.5 | 67.8 | 70.9 | 68.7 | 69.0 | 28.8 | 17.1 | 22.4 | 17.4 | 33.8 | 20.4 | 23.3 |
| • SAR (Dyn. LN) | 71.1 | 67.5 | 67.4 | 67.4 | 70.6 | 68.6 | 68.8 | 28.5 | 16.6 | 22.4 | 17.4 | 33.7 | 20.2 | 23.1 |
| • READ (Dyn. LN) | 71.3 | 68.5 | 68.5 | 68.4 | 71.8 | 69.0 | 69.6 | 36.4 | 25.3 | 28.9 | 27.3 | 35.6 | 26.6 | 30.0 |
| Source (Stat. (LN&AF)) | 73.7 | 65.5 | 67.9 | 70.3 | 67.9 | 70.3 | 69.3 | 37.0 | 25.5 | 16.8 | 21.6 | 27.3 | 25.5 | 25.6 |
| • Tent (Dyn. LN) | 73.9 | 67.4 | 69.2 | 70.4 | 66.5 | 70.5 | 69.6 | 10.6 | 2.6 | 1.8 | 2.8 | 5.3 | 4.1 | 4.5 |
| • EATA (Dyn. LN) | 73.7 | 66.1 | 68.5 | 70.3 | 67.9 | 70.1 | 69.4 | 39.2 | 26.1 | 22.9 | 26.0 | 31.7 | 30.4 | 29.4 |
| • SAR (Dyn. LN) | 73.7 | 65.4 | 68.2 | 69.9 | 67.2 | 70.2 | 69.1 | 37.4 | 9.5 | 11.0 | 12.1 | 26.8 | 23.7 | 20.1 |
| • READ (SAF) | **74.1** | **69.0** | **69.7** | **71.1** | **71.8** | **70.7** | **71.1** | **40.4** | **28.9** | **26.6** | **30.9** | **36.7** | **30.6** | **32.4** |

- negligible performance gains in some cases, implying that the reliability bias problem cannot be simply addressed by adopting the widely-used TTA paradigm, *i.e.*, updating the parameters of normalization layers.

- The proposed confidence-aware loss could bring performance gain for both late fusion and attention-based fusion. Applying the proposed SAF strategy with the loss could guarantee noise-resistant thus learning reliable attention for fusion. In other words, our READ could significantly improve the robustness against the cross-modal reliability bias.

Table 3: Comparisons with SOTA methods on VGGSound-C benchmark with **corrupted video modality (severity level 5)**.

| Methods | Noise | | | Blur | | | | Weather | | | | Digital | | | | Avg. |
|---|---|---|---|---|---|---|---|---|---|---|---|---|---|---|---|---|
| | Gauss. | Shot | Impul. | Defoc. | Glass | Mot. | Zoom | Snow | Frost | Fog | Brit. | Contr. | Elas. | Pix. | JPEG | |
| Source ((Stat. LN) & LF) | 37.7 | 36.5 | 37.8 | 52.7 | 51.3 | 55.2 | 53.7 | 51.9 | 52.3 | 50.4 | 55.3 | 45.2 | 52.5 | 51.7 | 52.3 | 49.1 |
| • MM-TTA (Dyn. LN) | 7.1 | 7.3 | 7.3 | 44.8 | 41.5 | 48.0 | 45.5 | 27.4 | 23.5 | 30.5 | 46.9 | 24.2 | 40.3 | 40.7 | 45.7 | 32.0 |
| • Tent (Dyn. LN) | 7.6 | 6.8 | 7.2 | 53.1 | 52.1 | 55.5 | 54.5 | 52.6 | 32.7 | 16.0 | 55.9 | 16.6 | 52.6 | 54.2 | 53.1 | 38.0 |
| • EATA (Dyn. LN) | 37.7 | 36.5 | 37.7 | 53.2 | 52.3 | 56.0 | 54.4 | 52.4 | 52.9 | 51.0 | 55.0 | 45.2 | 53.5 | 52.3 | 52.7 | 49.5 |
| • SAR (Dyn. LN) | 37.7 | 36.4 | 37.7 | 52.8 | 51.5 | 55.5 | 53.9 | 51.9 | 52.5 | 50.4 | 55.4 | 44.8 | 52.7 | 51.8 | 52.3 | 49.2 |
| • READ (Dyn. LN) | 42.1 | 41.5 | 42.1 | 49.3 | 50.9 | 53.5 | 52.5 | 50.6 | 52.1 | 51.1 | 54.0 | 46.2 | 52.5 | 49.1 | 50.2 | 49.2 |
| Source (Stat. (LN&AF)) | 52.8 | 52.7 | 52.7 | 57.2 | 57.2 | 58.7 | 57.6 | 56.4 | 56.6 | 55.6 | 58.9 | 53.7 | 56.9 | 55.8 | 56.9 | 56.0 |
| • Tent (Dyn. LN) | 52.7 | 52.7 | 52.7 | 56.7 | 56.5 | 57.9 | 57.2 | 55.9 | 56.3 | 56.3 | 58.4 | 54.0 | 57.4 | 56.2 | 56.7 | 55.8 |
| • EATA (Dyn. LN) | 53.0 | 52.8 | 53.0 | 57.2 | 57.1 | 58.6 | 57.8 | 56.3 | 56.8 | 56.4 | 59.0 | 54.1 | 57.4 | 56.1 | 57.0 | 56.2 |
| • SAR (Dyn. LN) | 52.9 | 52.8 | 52.9 | 57.2 | 57.1 | 58.6 | 57.6 | 56.3 | 56.7 | 55.9 | 58.9 | 54.0 | 57.0 | 56.0 | 57.0 | 56.1 |
| • READ (SAF) | **53.6** | **53.6** | **53.5** | **57.9** | **57.7** | **59.4** | **58.8** | **57.2** | **57.8** | 55.0 | **59.9** | **55.2** | **58.6** | **57.1** | **57.9** | **56.9** |

## 4.3 ABLATION AND ANALYTIC STUDIES

In this section, all the experiments are performed under Kinetics50 datasets with either fog noise on video or traffic noise on audio at severity level 5 unless otherwise stated.

**Ablation studies.** To verify the importance of each design, we investigate the variants of the method in Table 4, where one could have the following observations. First, Tent using SAF cannot always improve the robustness (*e.g.*, 34.9 to 22.7), which could be boiled down to the noise-dominant predictions in multi-modal TTA as depicted in Fig. 1(d). At that time, updating the parameters of the cross-attention might make model

Table 4: Ablation studies on Kinetics50-C benckmark. Pink denote the default setting.

| Variants | Video-fog | Audio-traffic |
|---|---|---|
| Tent ((Stat. LN) & SAF) | 22.7 | 69.0 |
| Ours ((Stat. LN) & AF) | 50.9 | 67.4 |
| Ours ((Dyn. LN) & SAF) | 58.1 | 69.3 |
| Ours ((Stat. LN) & SAF) | 57.4 | 69.0 |

overfitting on noise. Second, the proposed loss could also improve the robustness of the existing TTA paradigm that mainly updates the normalization layers (*e.g.*, 34.9 to 50.9). Third, applying the normalization layer updating mechanism to our SAF could slightly improve the performance (*e.g.*, 57.4 to 58.1). Note that, considering the efficiency, we maintain "(Stat. LN) & SAF" as our default setting. Due to space limitation, more comprehensive results are moved to Appendix D.1.

**Superiority of Reliable Fusion at Test-time.** As mentioned in Introduction, some imbalance modality learning studies (Zhang et al., 2023) could alleviate the modality reliability problem by alerting the training process in the source domain. To verify the necessity of reliable fusion at test-time, we compare our method with the recently proposed SOTA method (QMF (Zhang et al., 2023)) *on all kinds of corruptions*. In short, QMF could estimate the confidence of each modality by adopting an elaborately designed training pipeline and such a paradigm could naturally applied to handle reliability bias. As shown in Fig. 4, our method could achieve robustness superiority against both audio and video modalities compared to QMF, although the latter alters the training process in the source domain.

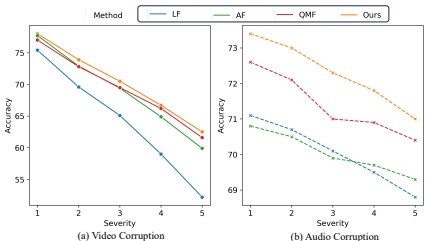

Figure 4: Comparisons between different fusion manners. In the figure, "LF", "AF", and "QMF" denote the variants of late fusion, vanilla attention-based fusion, and the training-time robust fusion of QMF, respectively.

**Reliable Fusion across Different Situations.** In this experiment, we further verify the claim in Introduction that the reliability bias cannot be completely eliminated by simply updating the normalization layers like existing TTA methods. From Fig. 3, one could observe that "Tent" would perform slightly robust fusion effects compared to "AF", which could be attributed to the narrowed domain gap by repurposing the LN. In contrast, our method shows remarkable improvements in the reliability estimation (attention value) on both video or audio bias situations with varying severities, which verifies the necessity of self-adaptive attention-based fusion paradigm for multi-modal TTA.

**Visualization on the Self-adaptive Attention.** The effectiveness of our SAF module could be verified from Fig. 5 with the following observations. First, adapting with clean/nearly-non-shifted test data, our SAF could maintain the importance between audio and video modalities (8.0 v.s. 41.5). Note that, the video is more informative for Kinetics50 under the action recognition task. Second, once the one modality is corrupted, SAF could make the model focus more on another rather reliable modalities, *e.g.*, from 8.0



Figure 5: Visualization on the self-adaptive attention. The blocks of the top left and bottom right denote the self-attention between audio and videos, respectively. The blocks of the top right and bottom left denote the cross-attention from audio to video and video to audio, respectively. The number upon the blocks denotes the mean of attention values across the adaptation process, which is amplified by $10,000$ times for clarity.

to 10.5. Meanwhile, the clean modality will reduce its attention to the corrupted one while the corrupted modality will increase the attention to the clean one, *e.g.*, from 30.2 to 23.6 and 3.7 to 5.1. Due to space limitations, we place more results in Appendix D.7.

## 5 CONCLUSIONS

In this paper, we formally study the multi-modal test-time adaptation. By delving into the distribution shifts in multi-modal scenarios, we reveal that the intra-modal distribution shifts will result in the information discrepancies across modalities, *i.e.*, the modality reliability bias challenge. To address this challenge, READ adopts the self-adaptive attention module to achieve reliable fusion across different modalities, and the confidence-aware loss function for robust adaptation. For comprehensive evaluations, we provide two new benchmarks with multi-modal reliability bias. In the future, we plan to explore more specific problems for multi-modal TTA with reliability bias under different applications and scenarios.

ACKNOWLEDGMENTS

This work was supported in part by NSFC under Grant U21B2040, 62176171; and in part by the Fundamental Research Funds for the Central Universities under Grant CJ202303.

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

# APPENDIX

## A  PROOFS TO THEOREMS

In this section, we present detailed proofs for Theorems 1 and 2 in the main paper. Without loss of generality, we omit the batch size and subscript of $p_i$, and obtain the form of $\mathcal{L}_{ra} = p \log{(e\gamma/p)}$.

**Theorem 1.** *The gradient direction produced by $\mathcal{L}_{ra}$ is non-monotonous.*

**Proof 1.** *The gradient of $\mathcal{L}_{ra}$ w.r.t. $p$ is in the form of*

$$\frac{\partial \mathcal{L}_{ra}}{\partial p} = \log{(\gamma)} - \log{(p)}. \tag{8}$$

*Clearly, $\partial \mathcal{L}_{ra}/\partial p > 0$ i.f.f. $p < \gamma$, and $\partial \mathcal{L}_{ra}/\partial p < 0$ i.f.f. $\gamma < p$, and $\partial \mathcal{L}_{ra}/\partial p = 0$ i.f.f. $p = \gamma$. Therefore, $\gamma$ is the stationary point of $\mathcal{L}_{ra}$ and $\mathcal{L}_{ra}$ is non-monotonous.*

$\square$

**Theorem 2.** *The gradient value will rise with increasing $p_i$ i.f.f. $p_i \in (\gamma, 1)$ or decreasing $p_i$ i.f.f. $p_i \in (0, \gamma)$.*

**Proof 2.** *The second-order gradient $\mathcal{L}_{ra}$ w.r.t. $p$ is in the form of*

$$\begin{aligned} \frac{\partial^2 \mathcal{L}_{ra}}{\partial p^2} &= \frac{\partial \frac{\partial \mathcal{L}_{ra}}{\partial p}}{\partial p} \\ &= -\frac{1}{p} < 0. \end{aligned} \tag{9}$$

*Therefore, $\partial \mathcal{L}_{ra}/\partial p$ decreases monotonically for $p \in (0, 1)$. Given that $\partial \mathcal{L}_{ra}/\partial p > 0$ if $p < \gamma$ and $\partial \mathcal{L}_{ra}/\partial p < 0$ if $\gamma < p$, $|\partial \mathcal{L}_{ra}/\partial p|$ will rise either with increasing $p_i$ ($p_i \in (\gamma, 1)$) or with decreasing $p_i$ ($p_i \in (0, \gamma)$).*

$\square$

## B  MORE DETAILS ABOUT THE BENCHMARKS

We construct two benchmarks for multi-modal TTA with reliability bias upon the VGGSound (Chen et al., 2020) and Kinetics (Kay et al., 2017) datasets. The two datasets are widely used for multi-modal event classification and action recognition. To be specific,

- VGGSound (Chen et al., 2020) is a large-scale video dataset that consists of 309 diverse classes and contains a broad spectrum of everyday audio events. All videos within the VGGSound dataset are "in the wild," meaning that they were captured in real-world settings, and the audio in the videos corresponds to the visual information, making the source of sound visually apparent. In other words, *the audio modality in the dataset will contain more task-specific information for the event classification task, compared to the visual modality*. Each video in this dataset has a fixed duration of 10 seconds. Due to the changes in video availability, we downloaded $14,046$ evaluation videos and thus obtain $14,046$ testing visual-audio pairs.

- Kinetics50 is a subset of Kinetics (Kay et al., 2017) dataset. Specifically, Kinetics is a comprehensive collection of YouTube videos, covering a diverse set of 400 distinct human action classes. Human actions depicted in these videos have been meticulously annotated through manual efforts employing Mechanical Turk. Due to the characteristic of action recognition, *video modality in the dataset will contain more information compared to the audio modality*. Additionally, all videos have been trimmed to a standardized duration of 10 seconds, centered around the specific action, ensuring consistency and relevance in the dataset. In our experiments, following Peng et al. (2022), we randomly select 50 classes from the dataset, obtaining the subset of Kinetics, *i.e.*, Kinetics50, with $29,204$ training pairs and $2,466$ test pairs.

To comprehensively evaluate modality bias, we introduce different distribution shifts on the video and audio modalities for the test sets of VGGSound (Chen et al., 2020) and Kinetics (Kay et al., 2017) datasets. For the video corruptions, we follow Hendrycks & Dietterich (2019) to apply 15 kinds of corruptions into the video, and each corruption is with 5 kinds of severity levels for extensive validations. Specifically, the corruptions on video modality include "Gaussian Noise", "Shot Noise", "Impulse Noise", "Defocus Blur", "Glass Blur", "Motion Blur", "Zoom Blur", "Snow", "Frost", "Fog", "Brightness", "Elastic", "Pixelate", "Contrast", and "JPEG". Similar to the video modality, we add 6 kinds of common audio noise[1] with 5 kind of severity levels captured in the wild. Specifically, the corruptions on audio modality include "Gaussian Noise", "Paris Traffic Noise", "Crowd Noise", "Rainy Noise", "Thunder Noise" and "Windy Noise". The case of the 15 video corruption types and 6 audio corruption types are visualized in Fig. 6 and Fig. 7,respectively. Finally, we obtain the corresponding corrupted benchmarks, named VGGSound-C and Kinetics50-C.

---

[1] https://freesound.org

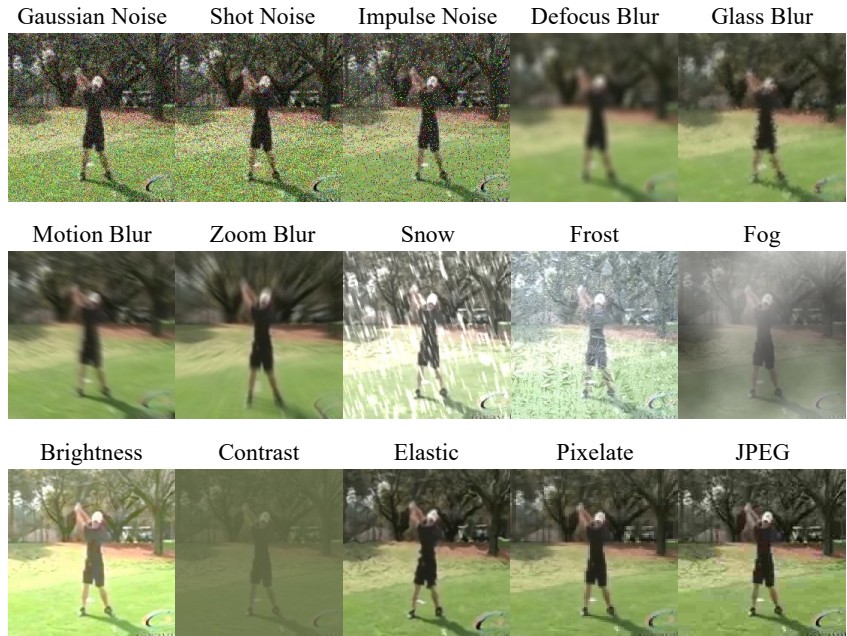

Figure 6: Visualization of various visual corruption types on the constructed Kinetics-C benchmark.

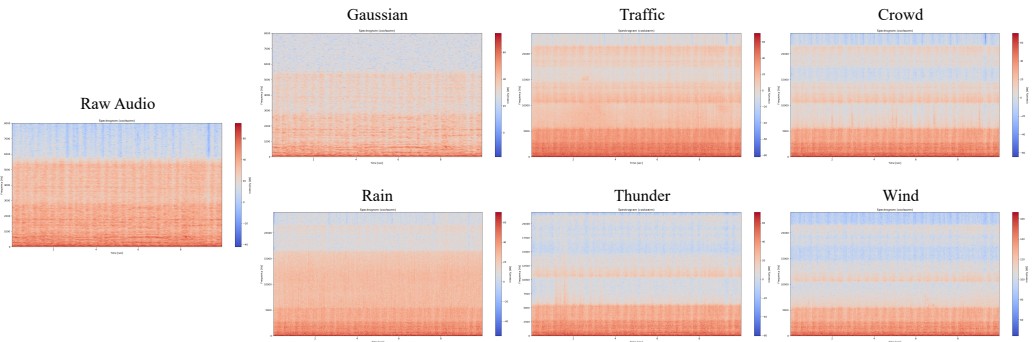

Figure 7: Mel spectrogram visualization of the raw audio and the corresponding audio corruption types on the constructed Kinetics-C benchmark.

## C    MORE DETAILS ABOUT THE BACKBONE

In the implementation, we use the CAV-MAE (Gong et al., 2023) model as the backbone. CAV-MAE adopts an encoder-decoder-like architecture that is pre-trained on large-scale video data with both the contrastive learning and mask image modeling paradigms. The CAV-MAE encoder consists of 11 Transformer layers dedicated to each modality for the modality-specific feature extraction, alongside one Transformer layer for cross-modal fusion. The input to the CAV-MAE encoder involves 10-second video clips containing both video and corresponding audio data. For the video stream, CAV-MAE samples 10 frames within each video clip and randomly selects one frame feeding into the visual Transformer encoder. For the audio stream, each 10-second audio waveform is converted into one spectrogram and then inputted to the audio Transformer encoder.

During the fine-tuning phase, we maintain the visual and audio encoders of the pre-trained model and add one randomly initialized classification head upon them. The fine-tuned model is regarded as the source model and denoted as "Source (Stat. (LN & AF))". Here, "Stat." is the short of "statical" that represents the frozen state of the layer normalization (LN) and attention-based fusion (AF) layers during the test-time phase. To investigate the robustness of different fusion manners, we design another variant of the source model that utilizes 12 Transformer layers for feature extraction and performs vanilla late fusion (LF) between the classification logits of each modality. The corresponding model variant is denoted as "Source ((Stat. LN) & LF)". During the test-time adaptation phase, unless otherwise specified, all baselines update the parameters of all normalization layers rooted in the source model, *i.e.*, referred to as "Dyn. LN" where "Dyn." is the short of "Dynamic". In contrast, as depicted in Fig. 2, our default approach in the READ framework involves updating only the parameters of the last Transformer layer (referred to as the AF layer) in a self-adaptive manner. We dub this paradigm as self-adaptive attention-based fusion, abbreviated as "SAF". SAF essentially repurposes the standard AF operation through modulating the parameters within the attention layer with the guidance of the proposed objective function (Eq. 7). As a result, the model would focus more on the unbiased modalities, leading to reliable cross-modal fusion during test time.

## D    MORE EXPERIMENT RESULTS

### D.1    MORE ABLATION RESULTS

We additionally present more comprehensive ablation results on Kinetics50 with both corrupted audio modality and video modality.

Table 5: Comprehensive ablation studies on Kinetics50-C with corrupted video modality.

| Methods | Noise | | | Blur | | | | Weather | | | | Digital | | | | Avg. |
|---|---|---|---|---|---|---|---|---|---|---|---|---|---|---|---|---|
| | Gauss. | Shot | Impul. | Defoc. | Glass | Mot. | Zoom | Snow | Frost | Fog | Brit. | Contr. | Elas. | Pix. | JPEG | |
| Tent ((Stat. LN) & SAF) | 45.3 | 45.7 | 45.1 | 66.6 | 58.1 | 70.5 | 65.8 | 60.8 | 57.2 | 22.7 | 75.2 | 48.6 | 66.1 | 63.7 | 53.4 | 56.3 |
| Ours ((Dyn. LN) & AF) | 47.8 | 48.2 | 47.6 | 67.7 | 64.2 | 71.0 | 68.2 | 63.9 | 62.6 | 50.9 | 75.3 | 53.3 | 66.9 | 67.8 | 63.8 | 61.3 |
| Ours ((Dyn. LN) & SAF) | 49.5 | 49.8 | 49.1 | 68.1 | 65.8 | 71.2 | 69.1 | 65.1 | 64.8 | 58.1 | 75.4 | 54.2 | 68.8 | 68.7 | 65.3 | 62.9 |
| Ours ((Stat. LN) & SAF) | 49.4 | 49.7 | 49.0 | 68.0 | 65.1 | 71.2 | 69.0 | 64.5 | 64.4 | 57.4 | 75.5 | 53.6 | 68.3 | 68.0 | 65.1 | 62.5 |

Table 6: Comprehensive ablation studies on Kinetics50-C with corrupted audio modality.

| Methods | Noise | | | Weather | | | Avg. |
|---|---|---|---|---|---|---|---|
| | Gauss. | Traff. | Crowd. | Rain | Thund. | Wind | |
| Tent ((Stat. LN) & SAF) | 73.7 | 69.0 | 69.6 | 70.4 | 69.2 | 70.6 | 70.4 |
| Ours ((Dyn. LN) & AF) | 73.8 | 67.4 | 69.0 | 70.6 | 70.5 | 70.3 | 70.3 |
| Ours ((Dyn. LN) & SAF) | 73.9 | 69.3 | 69.8 | 71.1 | 72.2 | 71.1 | 71.2 |
| Ours ((Stat. LN) & SAF) | 74.1 | 69.0 | 69.7 | 71.1 | 71.8 | 70.7 | 71.1 |

### D.2    RESULTS ON THE MIXED SEVERITY SETTING

To further investigate the robustness of our READ, we conduct more experiments on the Kinetics50-C benckmark under the settings of mixed severity, comparing with Tent (Wang et al., 2021) and SAR (Niu et al., 2023). To be specific, we create test pairs for each corruption type by blending severity levels from 1 to 5, resulting in 5N test pairs, where N represents the original size of the test data. After that, we shuffle the obtained test pairs and randomly choose N pairs for each corruption type. The results are depicted in Fig. 8, indicating the effectiveness of READ in addressing cross-modal reliability bias across various corruption types exhibiting mixed severity levels.

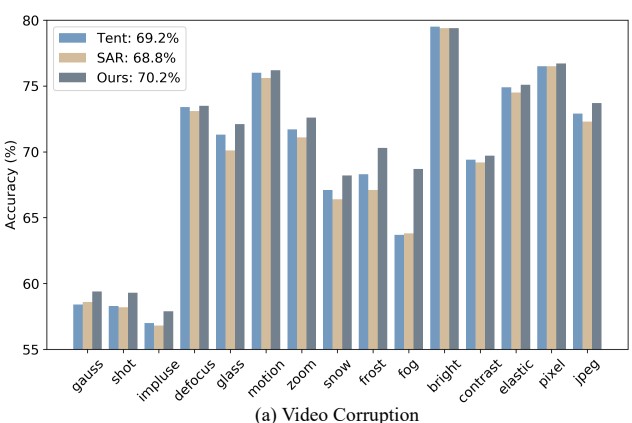

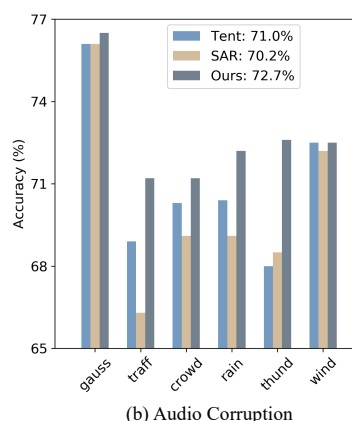

(a) Video Corruption                                            (b) Audio Corruption

Figure 8: Performance comparison among Tent, SAR, and our READ on the Kinetics50-C benchmark under mixed severity levels. The legend key provides an overview of the average performance of each approach across various corruption types.

### D.3    RESULTS ON THE MIXED DISTRIBUTION SHIFTS

We explore the efficacy of our READ approach in a more challenging scenario, *i.e.*, mixed distribution shifts, which is in line with the continual TTA setting (Gan et al., 2023; Wang et al., 2022). In this setting, both baseline methods (Tent and SAR) alongside our READ continually adapt to evolving corruption types, and the averaged performance across all corruption types are reported.

To ensure comprehensive evaluations, we vary the severity levels from 1 to 5. The results are summarized in Fig. 9. Although READ is not dedicatedly designed for the mixed distribution shifts challenge, it still achieve remarkable robustness. The results underscores the adaptability and resilience of READ.

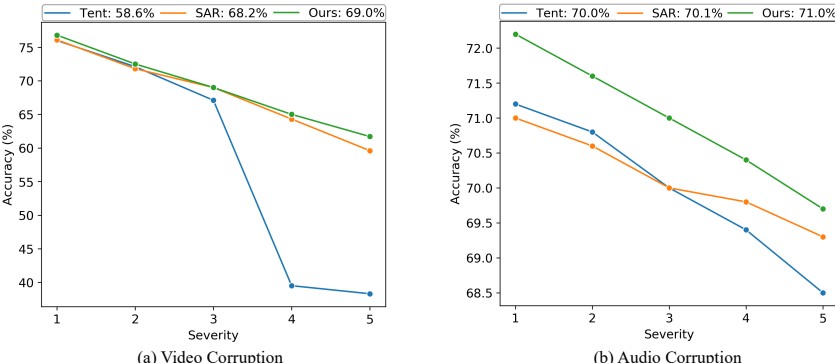

(a) Video Corruption        (b) Audio Corruption

Figure 9: Performance comparison among Tent, SAR, and our READ on the Kinetics50-C benchmark with mixed corruption types. The legend key provides an overview of the average performance of different severity levels.

## D.4 INFLUENCE OF THE HYPER-PARAMETER

In this section, we investigate the influence of the only hyper-parameter (*i.e.,* threshold $\gamma$ in Eq. 6) in our approach. To this end, we vary $\gamma$ in the range of $[0.1, 0.2, 0.3, e^{-1}, 0.4, 0.5]$ and perform corresponding experiments on the Kinetics50-C benchmark with fog and traffic corruptions. The results on Fig. 10 illustrates the stability of READ across varying threshold values of $\gamma$.

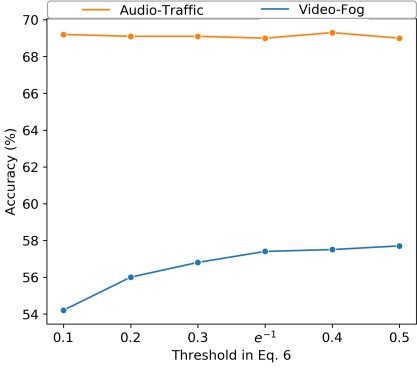

Figure 10: Sensitiveness analysis of our READ against the hyper-parameter on the Kinetics50-C benchmark with fog and traffic corruptions.

## D.5 EFFICIENCY COMPARISONS

Different from most TTA methods that updates the parameters of normalization layers, our READ repurpose last one Transformer layer of CAV-MAE (Gong et al., 2023) model as elaborated in Section 3. In this section, we compare the efficiency of the two paradigms. To this end, we choose the attention-fusion-based CAV-MAE model as source model (*i.e.*, source (Stat. (LN & AF))), and conduct experiments on the VGGSound-C benckmark. We measure both the size of learnable parameters and the GPU time during the test-time adaptation phase. Table 7 highlights that our READ accomplishes adaptation in less time. The efficiency of READ can be attributed to its module repurposing approach. Although the normalization layer updating scheme occupies fewer parameters, it demands more time for propagation.

Table 7: Efficiency comparisons among different approaches on the VGGSound-C benchmark.

| Method | #params (M) | GPU time (14,046 pairs) |
|---|---|---|
| Tent (Dyn. LN) | 0.2 | 209.5 seconds |
| EATA (Dyn. LN) | 0.2 | 207.6 seconds |
| SAR (Dyn. LN) | 0.2 | 286.1 seconds |
| READ (SAF) | 1.8 | 134.1 seconds |

## D.6 DIFFERENT MODULE REPURPOSE SCHEMES

In our default approach, we update $W_{\Theta^h}$ and $B_{\Theta^h}$ ($h \in Q, K, V$) within the last Transformer layer of the source model to ensure reliable fusion. This section explores the impact of different repurposing schemes. To this end, we design three variants: one that updates only the query and key projection layers, another that updates only the value projection layers, and a third that updates the final classification head. Table 8 illustrates that the default setting, updating the query, key, and value projection layers simultaneously, exhibits significant performance superiority. Modulating the classification head demonstrates minimal effectiveness (e.g., from $46.7$ to $49.1$). In contrast, the attention modulation scheme achieves adaptive fusion between discrepant modalities, mitigating the multi-modal reliability bias problem (e.g., from $46.7$ to $51.7$). Moreover, modulation on the query, key, and value projection layers introduces additional parameters for reliable fusion, resulting in further improvements in robustness (e.g., from $46.7$ to $57.4$).

Table 8: Comparisons between different modulation schemes on the Kinetics50-C benchmark with severity level of $5$.

| Corruption | Source | QK | V | MLP | QKV (ours) |
|---|---|---|---|---|---|
| Video-Fog | 46.7 | 51.7 | 53.6 | 49.1 | 57.4 |
| Audio-Traffic | 65.5 | 68.8 | 67.2 | 66.7 | 69.0 |

## D.7 MORE VISUALIZATION RESULTS ON THE ATTENTION MATRIX

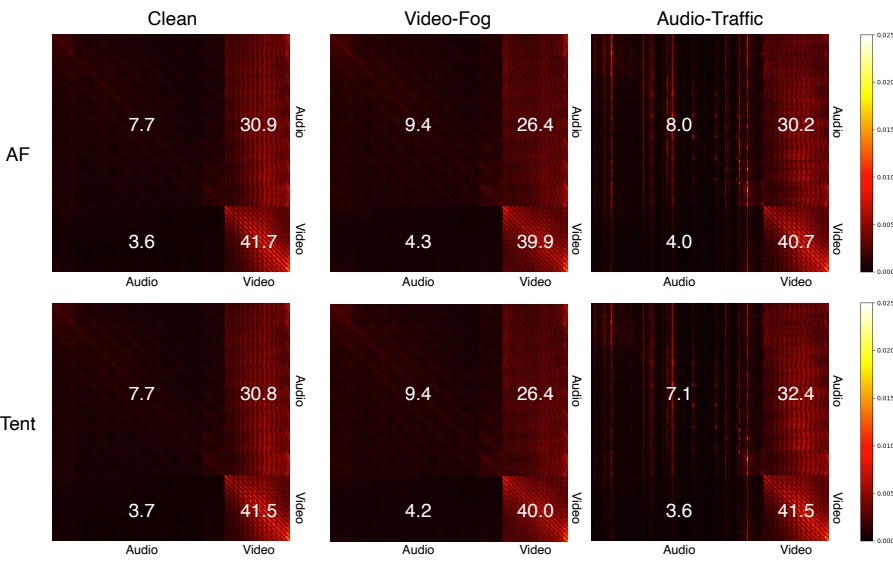

Figure 11: Visualization on the attention matrix of source model with vanilla attention-based fusion (AF), and the model adapted by Tent (Wang et al., 2021) with dynamic LN.

## D.8 MORE COMPARISON RESULTS ON DIFFERENT SEVERITY LEVELS

Table 9: Performance of different approaches on VGGSound and Kinetics50-C benchmark without any corruptions.

| Method | Source (Stat. (LN&AF)) | Tent | EATA | SAR | READ |
|---|---|---|---|---|---|
| VGGSound | 63.3 | 62.6 | 63.1 | 63.1 | 63.5 |
| Kinetics50 | 82.3 | 82.1 | 82.3 | 82.3 | 82.2 |

Table 10: Comparisons with SOTA methods on Kinetics50-C benchmark with **corrupted video modality (severity level 3)**.

| Methods | Noise | | | Blur | | | | Weather | | | | Digital | | | | Avg. |
|---|---|---|---|---|---|---|---|---|---|---|---|---|---|---|---|---|
| | Gauss. | Shot | Impul. | Defoc. | Glass | Mot. | Zoom | Snow | Frost | Fog | Brit. | Contr. | Elas. | Pix. | JPEG | |
| Source ((Stat. LN) & LF) | 46.6 | 47.8 | 46.9 | 71.0 | 63.4 | 74.4 | 68.1 | 62.1 | 58.9 | 65.4 | 77.6 | 68.2 | 76.1 | 77.1 | 73.0 | 65.1 |
| • MM-TTA (Dyn. LN) | 48.8 | 50.8 | 50.6 | 66.0 | 60.6 | 70.9 | 63.5 | 59.8 | 56.3 | 58.1 | 75.1 | 59.3 | 72.2 | 74.7 | 68.7 | 62.4 |
| • Tent (Dyn. LN) | 44.6 | 46.6 | 44.9 | 71.2 | 64.6 | 74.6 | 68.7 | 62.3 | 56.5 | 65.2 | 77.9 | 68.5 | 76.3 | 77.0 | 73.2 | 64.8 |
| • EATA (Dyn. LN) | 46.8 | 48.2 | 47.3 | 70.8 | 63.9 | 74.6 | 68.4 | 62.3 | 58.9 | 65.4 | 77.8 | 68.1 | 76.0 | 77.0 | 73.0 | 65.2 |
| • SAR (Dyn. LN) | 46.7 | 47.9 | 47.0 | 70.6 | 63.3 | 74.4 | 68.2 | 62.3 | 58.9 | 65.2 | 77.7 | 68.0 | 76.0 | 77.0 | 72.7 | 65.1 |
| • READ (Dyn. LN) | 49.3 | 50.0 | 49.4 | 71.1 | 65.7 | 75.0 | 70.3 | 64.5 | 61.5 | 67.1 | 78.1 | 69.5 | 76.6 | 77.2 | 73.7 | 66.6 |
| Source (Stat. (LN&AF)) | 54.1 | 54.8 | 54.6 | 73.5 | 68.3 | **76.6** | 71.5 | 69.2 | 64.7 | 69.5 | 79.3 | 72.1 | 77.6 | 79.4 | 75.4 | 69.4 |
| • Tent (Dyn. LN) | 54.2 | 55.1 | 55.2 | 73.6 | 69.6 | 76.8 | 71.9 | 69.5 | 65.6 | 70.2 | 79.4 | 72.9 | **78.3** | 79.2 | 75.3 | 69.8 |
| • EATA (Dyn. LN) | 54.4 | 54.9 | 55.0 | 73.4 | 69.1 | 76.5 | 71.6 | 69.2 | 65.1 | 69.5 | **79.5** | 72.3 | 77.7 | 79.1 | 75.2 | 69.5 |
| • SAR (Dyn. LN) | 54.2 | 54.8 | 55.0 | 73.1 | 68.2 | 76.4 | 71.1 | 69.1 | 64.8 | 69.4 | 79.1 | 72.0 | 77.4 | 79.1 | 75.0 | 69.2 |
| • READ (SAF) | **56.1** | **56.9** | **56.4** | **73.9** | **70.5** | 76.6 | **72.8** | **70.0** | **68.1** | **70.8** | 79.3 | **73.3** | 78.2 | **79.6** | **75.6** | **70.5** |

Table 11: Comparisons with SOTA methods on Kinetics50-C (left part) and VGGSound-C (right part) benchmarks with **corrupted audio modality (severity level 3)**.

| Methods | Noise | | | Weather | | | Avg. | Noise | | | Weather | | | Avg. |
|---|---|---|---|---|---|---|---|---|---|---|---|---|---|---|
| | Gauss. | Traff. | Crowd. | Rain | Thund. | Wind | | Gauss. | Traff. | Crowd. | Rain | Thund. | Wind | |
| Source ((Stat. LN) & LF) | 74.2 | 68.8 | 68.7 | 66.7 | 71.6 | 70.4 | 70.1 | 39.6 | 23.8 | 25.0 | 28.7 | 36.5 | 26.9 | 30.1 |
| • MM-TTA (Dyn. LN) | 72.8 | 69.6 | 68.9 | 68.7 | 70.7 | 70.3 | 70.2 | 13.8 | 7.1 | 7.6 | 16.2 | 10.6 | 5.4 | 10.1 |
| • Tent (Dyn. LN) | 74.2 | 69.0 | 69.6 | 64.8 | 71.9 | 71.1 | 70.1 | 11.2 | 4.1 | 3.4 | 5.2 | 12.8 | 5.1 | 7.0 |
| • EATA (Dyn. LN) | 74.1 | 68.8 | 69.1 | 67.3 | 71.8 | 70.6 | 70.3 | 40.3 | 23.9 | 24.7 | 28.7 | 36.5 | 26.9 | 30.2 |
| • SAR (Dyn. LN) | 73.9 | 68.8 | 68.9 | 66.7 | 71.6 | 70.3 | 70.0 | 39.9 | 23.6 | 24.9 | 28.7 | 36.4 | 26.8 | 30.0 |
| • READ (Dyn. LN) | 74.2 | 69.6 | 70.0 | 69.0 | 72.7 | 70.8 | 71.0 | 44.5 | 29.9 | 31.5 | 33.2 | 37.0 | 31.2 | 34.6 |
| Source (Stat. (LN&AF)) | 75.9 | 64.4 | 68.7 | 70.3 | 67.9 | 70.3 | 69.3 | 42.1 | 29.4 | 19.5 | 27.6 | 31.2 | 29.4 | 29.9 |
| • Tent (Dyn. LN) | 73.9 | 67.4 | 69.2 | 69.3 | 69.0 | 72.1 | 70.1 | 8.1 | 4.0 | 2.3 | 4.7 | 7.8 | 6.1 | 5.5 |
| • EATA (Dyn. LN) | 76.0 | 65.7 | 68.9 | 69.8 | 69.1 | 72.1 | 70.3 | 46.7 | 30.5 | 28.0 | 31.4 | 35.4 | **33.8** | 34.3 |
| • SAR (Dyn. LN) | 76.0 | 64.6 | 68.7 | 69.3 | 68.6 | 72.2 | 69.9 | 43.1 | 17.3 | 8.3 | 29.0 | 31.6 | 30.5 | 26.6 |
| • READ (SAF) | **76.4** | **69.6** | **70.8** | **72.0** | **72.6** | **72.3** | **72.3** | **47.3** | **32.7** | **29.9** | **33.2** | **38.3** | 33.7 | **35.8** |

Table 12: Comparisons with SOTA methods on VGGSound-C benchmark with **corrupted video modality (severity level 3)**.

| Methods | Noise | | | Blur | | | | Weather | | | | Digital | | | | Avg. |
|---|---|---|---|---|---|---|---|---|---|---|---|---|---|---|---|---|
| | Gauss. | Shot | Impul. | Defoc. | Glass | Mot. | Zoom | Snow | Frost | Fog | Brit. | Contr. | Elas. | Pix. | JPEG | |
| Source ((Stat. LN) & LF) | 45.6 | 45.3 | 45.4 | 55.7 | 54.0 | 57.6 | 55.4 | 55.1 | 53.7 | 53.4 | 58.5 | 53.9 | 58.3 | 58.1 | 56.5 | 53.8 |
| • MM-TTA (Dyn. LN) | 18.6 | 17.5 | 15.8 | 50.4 | 44.3 | 51.8 | 48.4 | 41.4 | 28.1 | 46.5 | 52.0 | 46.2 | 52.0 | 52.0 | 51.6 | 41.1 |
| • Tent (Dyn. LN) | 19.8 | 17.2 | 18.4 | 55.9 | 55.3 | 57.3 | 55.9 | 55.3 | 45.3 | 34.8 | 58.4 | 56.4 | 58.4 | 58.4 | 57.1 | 46.9 |
| • EATA (Dyn. LN) | 45.8 | 45.6 | 45.7 | 56.3 | 55.2 | 58.0 | 56.0 | 55.8 | 54.4 | 54.5 | 58.9 | 55.3 | 58.8 | 58.5 | 57.1 | 54.4 |
| • SAR (Dyn. LN) | 45.4 | 45.2 | 45.2 | 55.8 | 54.3 | 57.7 | 55.6 | 55.3 | 53.9 | 53.7 | 58.5 | 54.2 | 58.5 | 58.2 | 56.7 | 53.9 |
| • READ (Dyn. LN) | 46.0 | 46.0 | 46.3 | 53.0 | 52.9 | 56.3 | 54.1 | 53.8 | 53.3 | 53.0 | 58.0 | 53.8 | 57.7 | 56.8 | 55.1 | 53.1 |
| Source (Stat. (LN & AF)) | 54.7 | 54.6 | 54.7 | 59.3 | 58.4 | 60.4 | 59.0 | 58.3 | 57.4 | 57.8 | 61.3 | 58.0 | 61.0 | 60.9 | 60.0 | 58.4 |
| • Tent (Dyn. LN) | 54.4 | 54.3 | 54.4 | 58.8 | 57.8 | 59.7 | 58.5 | 57.6 | 56.9 | 58.6 | 60.6 | 58.0 | 61.0 | 60.9 | 60.0 | 58.1 |
| • EATA (Dyn. LN) | 54.8 | 54.6 | 54.8 | 59.4 | 58.4 | 60.3 | 59.1 | 58.2 | 57.5 | 58.7 | 61.3 | 58.9 | 61.1 | 60.8 | 60.1 | 58.5 |
| • SAR (Dyn. LN) | 54.8 | 54.6 | 54.7 | 59.4 | 58.3 | 60.3 | 58.9 | 58.3 | 57.5 | 58.2 | 61.2 | 58.3 | 60.9 | 60.8 | 59.9 | 58.4 |
| • READ (SAF) | **55.3** | **55.4** | **55.4** | **60.0** | **59.1** | **61.1** | **59.8** | **59.2** | **58.5** | **59.3** | **61.9** | **59.8** | **61.5** | **61.5** | **60.7** | **59.2** |

