# OpenReview forum: "Test-time Adaptation against Multi-modal Reliability Bias"
_ICLR.cc/2024/Conference — ICLR 2024 poster_

### Official Review · Reviewer_kzfx · 2023-10-28

**Soundness:** 3 good
**Presentation:** 4 excellent
**Contribution:** 4 excellent
**Rating:** 8
**Confidence:** 5

**Summary:**

This paper studies the problem of multi-modal test-time adaption (TTA) under the challenge of reliability bias, which refers to the information discrepancies across different modalities due to the distribution shifts between domains. The paper proposes a novel method, RFRA, which consists of two modules: a self-adaptive attention module for reliable fusion across modalities, and a confidence-aware loss function for robust adaption. The paper also provides two new benchmarks for multi-modal TTA with reliability bias based on Kinetics and VGGSound datasets. The paper shows that RFRA outperforms several state-of-the-art TTA methods on these benchmarks under various corruptions.

**Strengths:**

1. This paper is well-written, well-organized, and easy to follow.
2. The paper addresses a novel and important problem, i.e.,  multi-modal TTA with reliability bias, which has not been well-studied in the literature.  Accordingly, the paper proposes an effective method, RFRA, which leverages self-adaptive attention and confidence-aware loss to achieve reliable fusion and robust adaption across modalities.   Moreover, the confidence-aware loss is simple but effective and enjoy the non-monotonous gradient property.
3. The paper provides two new benchmarks for multi-modal TTA with reliability bias, which could facilitate future research on this topic. The paper conducts extensive experiments and ablation studies on these benchmarks to demonstrate the effectiveness and superiority of RFRA over existing TTA methods.

**Weaknesses:**

1. The paper lacks some experimental details, such as clarifying the terms “Statical” and “Dynamic” in Table 1 and explaining the distinctions between the proposed Self-Adaptive Attention and traditional self-attention. It is unclear what these terms mean and how they affect the performance of different methods. The paper should provide more definitions and discussions on these terms. Moreover, the paper should elaborate on how the Self-Adaptive Attention differs from the conventional self-attention in terms of design, implementation, and advantages.
2. The confidence threshold $gamma$ work as the important parameter in the confidence-aware loss.  However, the influence of the confidence threshold $\gamma$ on TTA performance is not explored, and it would be beneficial to understand its role more explicitly. The paper could conduct more experiments and analysis to show how different values of $\gamma$ affect the accuracy and robustness of TTA. The paper could provide provide some insights on how to choose an appropriate value of $\gamma$.
3. How does the attention value in Figure 4 calculated? Some explanation is needed. The paper should provide more details on how to compute the attention value for each modality pair in Figure 4.
4. An outlying problem, how does the audio information used in autonomous vehicle? As most scenarios of autonomous vehicle might use the visual sensor and could your method used in such case?
5. There is a typographical error in Eq. (4); “A” should be bold. This is a minor mistake that can be easily corrected.

**Questions:**

The primary questions for the rebuttal primarily arise from the "weaknesses" section. It would be highly appreciated if the authors could provide further explanations regarding the experiments and address the raised concerns, which will strengthen the paper. Overall, I recommend accepting this paper.

---

> ### Author Response · Authors · 2023-11-19
> **The Response to Reviewer kzfx (Part 1)**
>
> Thanks for the detailed comments. In the following, we will answer your questions one by one.
>
> > *Q1.1: The paper lacks some experimental details, such as **clarifying the terms “Statical” and “Dynamic”** in Table 1 and explaining the distinctions between the proposed Self-Adaptive Attention and traditional self-attention. It is unclear what these terms mean and how they affect the performance of different methods. The paper should provide more definitions and discussions on these terms.* Moreover, the paper should **elaborate on how the Self-Adaptive Attention differs from the conventional self-attention** in terms of design, implementation, and advantages.*
>
> **A1.1:** We apologize for any confusion regarding these terms. Allow us to elaborate:
>
> **"Statical" and "Dynamic"**. "Statical" denotes the state where the layer normalization (LN) and attention-based fusion (AF) layers remain frozen during the test-time phase. Conversely, "Dynamic" implies the updating of parameters within all normalization layers rooted in the source model as most existing TTA methods do in the test-time phase.
>
> **Self-Adaptive Attention v.s. traditional self-attention**. The traditional self-attention mechanism usually keeps the parameters inherited from the source model and performs information integration across modalities. Apparently, the distribution shift between training and test-time data might hinder the similarity estimation between tokens. As a result, reliable fusion on biased modalities cannot be guaranteed. Instead, we hope that the model could focus more on the unbiased modalities and avoid the interoperation from the bias. To this end, we propose repurposing the cross-modal attention-based fusion layers in a self-adaptive way. As depicted in Figure 2 of the manuscript, our default approach in the RFRA framework involves updating only the parameters of the last Transformer layer (referred to as the AF layer) in a self-adaptive manner. We dub this paradigm as self-adaptive attention-based fusion, abbreviated as "SAF". SAF essentially repurposes the standard AF operation by modulating the parameters within the attention layer with the guidance of the proposed objective function.
>
> > *Q2: The confidence threshold gamma work as the important parameter in the confidence-aware loss. However, the influence of the confidence threshold $\gamma$ on TTA performance is not explored, and it would be beneficial to understand its role more explicitly. The paper could conduct more experiments and analysis to **show how different values of $\gamma$ affect the accuracy and robustness of TTA**. The paper could provide provide some insights on how to choose an appropriate value of $\gamma$.*
>
> **A2:** Thanks for your comment. In response to your concern, we investigate the influence of the only hyper-parameter (\textit{i.e.,} threshold $\gamma$ in Eq. 6) in our approach. To this end, we vary $\gamma$ in the range of $[0.1, 0.2, 0.3, e^{-1}, 0.4, 0.5]$ and perform corresponding experiments on the Kinetics50-C benchmark with fog and traffic corruptions. Results are depicted in Fig. 10 within the revised manuscript. For your convenience, we attach the corresponding numerical results in the following table.
>
> | Threshold ($\gamma$ in Eq. 6) | 0.1  | 0.2  | 0.3  | $e^{-1}$ | 0.4  | 0.5  |
> | :---------------------------- | :--: | :--: | :--: | :------: | :--: | :--: |
> | Video-Fog                     | 54.2 |  56  | 56.8 |   57.4   | 57.5 | 57.7 |
> | Audio-Traffic                 | 69.2 | 69.1 | 69.1 |   69.0   | 69.3 | 69.0 |
>
> The results illustrate the stability of RFRA across varying threshold values of $\gamma$.
>
> > *Q3: How does the attention value in Figure 4 calculated? Some explanation is needed. The paper should provide more **details on how to compute the attention value for each modality pair in Figure 4**.*
>
> **A3:** Thanks for your comment. In Figure 4, the attention values between modalities, denoted as "Attention X-Y" (X, Y $\in$ {A, V}), are calculated using Equation 4 from the manuscript:
>
> $$\mathbf{A}=\operatorname{Softmax}\left(\frac{\mathbf{Q} \mathbf{K}^T}{\sqrt{d}}\right)$$.
>
> For instance, "Attention X-X" represents the self-attention, computed by setting both the query ($\mathbf{Q}$) and key ($\mathbf{K}$) as the tokens of modality X. Meanwhile, "Attention X-Y" (X, Y $\in$ {A, V}) corresponds to the configuration where the query and key tokens come from modality X and modality Y, respectively.

---

> > ### Author Response · Authors · 2023-11-19
> > **The Response to Reviewer kzfx (Part 2)**
> >
> > > *Q4: An outlying problem, **how does the audio information used in autonomous vehicle**? As most scenarios of autonomous vehicle might use the visual sensor and could your method used in such case?*
> >
> > **A4:** Thanks for your comment. It's crucial to note that the multi-modal reliability bias revealed in this study includes but not limited to autonomous vehicles and can manifest across various modalities in diverse scenarios. While the majority of autonomous vehicles primarily incorporate sensors like cameras, radars, and lidars, the potential utilization of microphones and audio data in autonomous driving has received increasing attention. Interestingly, audio information could significantly complement the capabilities of the autonomous system. For instance, as highlighted in [A], audio input aids in emergency vehicle recognition. This underlines the broader applicability of the multi-modal reliability bias challenge and its potential implications for the wider community.
> >
> > > *Q5: There is a **typographical error** in Eq. (4); “A” should be bold. This is a minor mistake that can be easily corrected.*
> >
> > **A5:** We appreciate your feedback on the notation typos. In the updated version, we have carefully revised the typos.
> >
> > [A] Ivan Kharitonov. "Survey on Acoustic Sensors in Self-Driving Cars." 2023, https://hackernoon.com/survey-on-acoustic-sensors-in-self-driving-cars](https://hackernoon.com/survey-on-acoustic-sensors-in-self-driving-cars)

---

### Official Review · Reviewer_9FmQ · 2023-10-28

**Soundness:** 4 excellent
**Presentation:** 3 good
**Contribution:** 4 excellent
**Rating:** 8
**Confidence:** 5

**Summary:**

This paper introduces a novel approach for addressing reliability bias in multi-modal test-time adaptation (TTA), a challenge arising from information disparities between modalities due to distribution shifts. To investigate the impact of reliability bias, the authors conduct comprehensive analyses involving various multi-modal fusion strategies and state-of-the-art TTA methods. The results underscore two pivotal aspects of effective TTA against reliability bias: dynamic information integration across modalities and noise-resilient adaptation across domains. To tackle these challenges, the authors devise a self-adaptive attention module to facilitate reliable cross-modal fusion and a confidence-aware loss function to ensure robustness against noisy predictions. Furthermore, this paper contributes two benchmark datasets focusing on multi-modal action recognition and event classification. Extensive comparison experiments against existing TTA methods and imbalanced multi-modal learning methods validate the effectiveness of the proposed method.

**Strengths:**

1. This paper studies a new challenge (i.e., reliability bias) for multi-modal test-time adaption. Test-time adaption methods aim at adopting the pre-trained model from the source domain to the target domain in real-time and most existing of them focus on single-modality tasks against domain shifts. On the one hand, this paper takes the more complex multi-modal scenarios into consideration. On the other hand, the authors study and tackle the reliability bias challenge.

2. The authors conduct extensive experiments to validate the importance of developing robust TTA methods against reliability bias. On the one hand, the existing cross-modal fusion methods (late fusion, attention-based fusion, etc.) would suffer from reliability bias and cannot achieve reliable cross-modal fusion. On the other hand, the existing TTA method cannot completely reconcile the distribution shifts by updating the parameters of normalization layers, leading to surviving reliability bias across modalities. Furthermore, the authors show that simply handling reliability bias during test time takes more superiority compared to the imbalanced multi-modal learning methods that alter the training process to handle the problem.

3. The proposed method is novel and technically sound. First, the authors focus on the characteristics of multi-modal TTA and design the self-adaptive attention module that repurposes the attention layers during test time for achieving reliable cross-modal fusion. I think the design would inspire the community to design task-specific parameter modulation instead of solely updating the parameters of normalization layers following most existing methods. Second, to achieve robustness against heavy noise during adaption, the authors propose the robust loss function which not only eliminates the influence of noisy predictions but also boosts utilization of the clean predictions with theoretical guarantees.

**Weaknesses:**

Although this paper is well-motivated and extensively validated, I still have the following concerns or suggestions, hoping to make the paper more clear and solid.

1. The experiment results are mainly obtained under the setting of severity 5. To establish the method's generality, it is encouraged to expand the empirical results across a spectrum of scenarios, including different severity levels. A broader array of experiments, encompassing various severity levels, would not only fortify the method's reliability but also enhance the comprehensiveness of this study. Moreover,  it would be beneficial to evaluate how the method performs in the context of test-time adaptation (TTA) under unbiased reliability conditions. Specifically, investigating the method's effectiveness in both (i.d.d. and non-i.d.d. scenarios would render it more practical and versatile.
2. The paper contains extensive analysis and experiments, but some settings require further clarification. For instance, it's not entirely clear what "Attention A-V" means in Figure 4 and how the results demonstrate the method's robustness. In the analysis of Figure 5, claims are made about the importance of maintenance between audio and video modalities, but the contrast with clean results is not evident. Additional clarification or supplementary results are needed to support these claims.
3. There are a few typos and vague statements in the paper, such as "trafﬁc noise in the audio modality" in the caption of Figure 1, "information bias" on Page 2, and inconsistent notations like "(Stat. LN) & AF" or "Stat. (LN & AF)" in Tables 1-3. These should be corrected for clarity and consistency.

**Questions:**

My questions mainly lie in some unclear experiment analysis and  the generalizability of the proposed approach to a broader range of severity levels.

---

> ### Author Response · Authors · 2023-11-19
> **The Response to Reviewer 9FmQ (Part 1)**
>
> Thanks for the insightful reviews. We will answer your questions one by one in the following.
>
> > *Q1.1: The experiment results are mainly obtained under the setting of severity 5. To establish the method's generality, it is encouraged to **expand the empirical results across a spectrum of scenarios, including different severity levels**. A broader array of experiments, encompassing various severity levels, would not only fortify the method's reliability but also enhance the comprehensiveness of this study.*
>
> **A1.1:** Thanks for your valuable suggestions. In the submission, we have reported the comparison results on VGGSound-C and Kinetics50-C benchmarks under 15 visual corruption and 6 audio corruption with a severity level of 5.
>
> In response to your constructive feedback, we conduct additional experiments to assess our RFRA's performance across different corruption types at severity level 3. We've summarized these results in Tables 10-12 within the revised manuscript. For your convenience, the corresponding results are provided in the following tables.
>
> | Kinetics50-C (Video Corruption, Severity Level 3) |  Gauss.  |   Shot   |  Impul.  |  Defoc.  |  Glass   |   Mot.   |   Zoom   |  Snow  | Frost    | Fog      | Brit.    | Digital  | Contr.   | Pix.     | JPEG     | AVG      |
> | :------------------------------------------------ | :------: | :------: | :------: | :------: | :------: | :------: | :------: | :----: | -------- | -------- | -------- | -------- | -------- | -------- | -------- | -------- |
> | Source ((Stat. LN) \& LF)                         |   46.6   |   47.8   |   46.9   |    71    |   63.4   |   74.4   |   68.1   |  62.1  | 58.9     | 65.4     | 77.6     | 68.2     | 76.1     | 77.1     | 73       | 65.1     |
> | + MM-TTA (Dyn. LN)                                |   48.8   |   50.8   |   50.6   |    66    |   60.6   |   70.9   |   63.5   |  59.8  | 56.3     | 58.1     | 75.1     | 59.3     | 72.2     | 74.7     | 68.7     | 62.4     |
> | + Tent (Dyn. LN)                                  |   44.6   |   46.6   |   44.9   |   71.2   |   64.6   |   74.6   |   68.7   |  62.3  | 56.5     | 65.2     | 77.9     | 68.5     | 76.3     | 77       | 73.2     | 64.8     |
> | + EATA (Dyn. LN)                                  |   46.8   |   48.2   |   47.3   |   70.8   |   63.9   |   74.6   |   68.4   |  62.3  | 58.9     | 65.4     | 77.8     | 68.1     | 76       | 77       | 73       | 65.2     |
> | + SAR (Dyn. LN)                                   |   46.7   |   47.9   |    47    |   70.6   |   63.3   |   74.4   |   68.2   |  62.3  | 58.9     | 65.2     | 77.7     | 68       | 76       | 77       | 72.7     | 65.1     |
> | + RFRA (Dyn. LN)                                  |   49.3   |    50    |   49.4   |   71.1   |   65.7   |    75    |   70.3   |  64.5  | 61.5     | 67.1     | 78.1     | 69.5     | 76.6     | 77.2     | 73.7     | 66.6     |
> | Source (Stat. (LN\&AF))                           |   54.1   |   54.8   |   54.6   |   73.5   |   68.3   | **76.6** |   71.5   |  69.2  | 64.7     | 69.5     | 79.3     | 72.1     | 77.6     | 79.4     | 75.4     | 69.4     |
> | + Tent (Dyn. LN)                                  |   54.2   |   55.1   |   55.2   |   73.6   |   69.6   |   76.8   |   71.9   |  69.5  | 65.6     | 70.2     | 79.4     | 72.9     | **78.3** | 79.2     | 75.3     | 69.8     |
> | + EATA (Dyn. LN)                                  |   54.4   |   54.9   |    55    |   73.4   |   69.1   |   76.5   |   71.6   |  69.2  | 65.1     | 69.5     | **79.5** | 72.3     | 77.7     | 79.1     | 75.2     | 69.5     |
> | + SAR (Dyn. LN)                                   |   54.2   |   54.8   |    55    |   73.1   |   68.2   |   76.4   |   71.1   |  69.1  | 64.8     | 69.4     | 79.1     | 72       | 77.4     | 79.1     | 75       | 69.2     |
> | + RFRA (SAF)                                      | **56.1** | **56.9** | **56.4** | **73.9** | **70.5** | **76.6** | **72.8** | **70** | **68.1** | **70.8** | 79.3     | **73.3** | 78.2     | **79.6** | **75.6** | **70.5** |

---

> > ### Author Response · Authors · 2023-11-19
> > **The Response to Reviewer 9FmQ (Part 2)**
> >
> > | VGGSound-C (Video Corruption, Severity Level 3) |  Gauss.  |   Shot   |  Impul.  | Defoc. |  Glass   |   Mot.   |   Zoom   |   Snow   | Frost    | Fog      | Brit.    | Digital  | Contr.   | Pix.     | JPEG     | AVG      |
> > | :---------------------------------------------- | :------: | :------: | :------: | :----: | :------: | :------: | :------: | :------: | -------- | -------- | -------- | -------- | -------- | -------- | -------- | -------- |
> > | Source ((Stat. LN) \& LF)                       |   45.6   |   45.3   |   45.4   |  55.7  |    54    |   57.6   |   55.4   |   55.1   | 53.7     | 53.4     | 58.5     | 53.9     | 58.3     | 58.1     | 56.5     | 53.8     |
> > | + MM-TTA (Dyn. LN)                              |   18.6   |   17.5   |   15.8   |  50.4  |   44.3   |   51.8   |   48.4   |   41.4   | 28.1     | 46.5     | 52       | 46.2     | 52       | 52       | 51.6     | 41.1     |
> > | + Tent (Dyn. LN)                                |   19.8   |   17.2   |   18.4   |  55.9  |   55.3   |   57.3   |   55.9   |   55.3   | 45.3     | 34.8     | 58.4     | 56.4     | 58.4     | 58.4     | 57.1     | 46.9     |
> > | + EATA (Dyn. LN)                                |   45.8   |   45.6   |   45.7   |  56.3  |   55.2   |    58    |    56    |   55.8   | 54.4     | 54.5     | 58.9     | 55.3     | 58.8     | 58.5     | 57.1     | 54.4     |
> > | + SAR (Dyn. LN)                                 |   45.4   |   45.2   |   45.2   |  55.8  |   54.3   |   57.7   |   55.6   |   55.3   | 53.9     | 53.7     | 58.5     | 54.2     | 58.5     | 58.2     | 56.7     | 53.9     |
> > | + RFRA (Dyn. LN)                                |    46    |    46    |   46.3   |   53   |   52.9   |   56.3   |   54.1   |   53.8   | 53.3     | 53       | 58       | 53.8     | 57.7     | 56.8     | 55.1     | 53.1     |
> > | Source (Stat. (LN\&AF))                         |   54.7   |   54.6   |   54.7   |  59.3  |   58.4   |   60.4   |    59    |   58.3   | 57.4     | 57.8     | 61.3     | 58       | 61       | 60.9     | 60       | 58.4     |
> > | + Tent (Dyn. LN)                                |   54.4   |   54.3   |   54.4   |  58.8  |   57.8   |   59.7   |   58.5   |   57.6   | 56.9     | 58.6     | 60.6     | 58       | 61       | 60.9     | 60       | 58.1     |
> > | + EATA (Dyn. LN)                                |   54.8   |   54.6   |   54.8   |  59.4  |   58.4   |   60.3   |   59.1   |   58.2   | 57.5     | 58.7     | 61.3     | 58.9     | 61.1     | 60.8     | 60.1     | 58.5     |
> > | + SAR (Dyn. LN)                                 |   54.8   |   54.6   |   54.7   |  59.4  |   58.3   |   60.3   |   58.9   |   58.3   | 57.5     | 58.2     | 61.2     | 58.3     | 60.9     | 60.8     | 59.9     | 58.4     |
> > | + RFRA (SAF)                                    | **55.3** | **55.4** | **55.4** | **60** | **59.1** | **61.1** | **59.8** | **59.2** | **58.5** | **59.3** | **61.9** | **59.8** | **61.5** | **61.5** | **60.7** | **59.2** |
> >
> > | Kinetics50-C (Audio Corruption, Severity Level 3) | Gauss.   | Traff.   | Crowd.   | Rain   | Thund.   | Wind     | Avg.     |
> > | ------------------------------------------------- | -------- | -------- | -------- | ------ | -------- | -------- | -------- |
> > | Source ((Stat. LN) \& LF)                         | 74.2     | 68.8     | 68.7     | 66.7   | 71.6     | 70.4     | 70.1     |
> > | + MM-TTA (Dyn. LN)                                | 72.8     | 69.6     | 68.9     | 68.7   | 70.7     | 70.3     | 70.2     |
> > | + Tent (Dyn. LN)                                  | 74.2     | 69       | 69.6     | 64.8   | 71.9     | 71.1     | 70.1     |
> > | + EATA (Dyn. LN)                                  | 74.1     | 68.8     | 69.1     | 67.3   | 71.8     | 70.6     | 70.3     |
> > | + SAR (Dyn. LN)                                   | 73.9     | 68.8     | 68.9     | 66.7   | 71.6     | 70.3     | 70       |
> > | + RFRA (Dyn. LN)                                  | 74.2     | 69.6     | 70       | 69     | 72.7     | 70.8     | 71       |
> > | Source (Stat. (LN\&AF))                           | 75.9     | 64.4     | 68.7     | 70.3   | 67.9     | 70.3     | 69.3     |
> > | + Tent (Dyn. LN)                                  | 73.9     | 67.4     | 69.2     | 69.3   | 69       | 72.1     | 70.1     |
> > | + EATA (Dyn. LN)                                  | 76       | 65.7     | 68.9     | 69.8   | 69.1     | 72.1     | 70.3     |
> > | + SAR (Dyn. LN)                                   | 76       | 64.6     | 68.7     | 69.3   | 68.6     | 72.2     | 69.9     |
> > | + RFRA (SAF)                                      | **76.4** | **69.6** | **70.8** | **72** | **72.6** | **72.3** | **72.3** |

---

> > > ### Author Response · Authors · 2023-11-19
> > > **The Response to Reviewer 9FmQ (Part 3)**
> > >
> > > | VGGSound-C (Audio Corruption, Severity Level 3) | Gauss.   | Traff.   | Crowd.   | Rain     | Thund.   | Wind     | Avg.     |
> > > | ----------------------------------------------- | -------- | -------- | -------- | -------- | -------- | -------- | -------- |
> > > | Source ((Stat. LN) \& LF)                       | 39.6     | 23.8     | 25       | 28.7     | 36.5     | 26.9     | 30.1     |
> > > | + MM-TTA (Dyn. LN)                              | 13.8     | 7.1      | 7.6      | 16.2     | 10.6     | 5.4      | 10.1     |
> > > | + Tent (Dyn. LN)                                | 11.2     | 4.1      | 3.4      | 5.2      | 12.8     | 5.1      | 7.0      |
> > > | + EATA (Dyn. LN)                                | 40.3     | 23.9     | 24.7     | 28.7     | 36.5     | 26.9     | 30.2     |
> > > | + SAR (Dyn. LN)                                 | 39.9     | 23.6     | 24.9     | 28.7     | 36.4     | 26.8     | 30       |
> > > | + RFRA (Dyn. LN)                                | 44.5     | 29.9     | 31.5     | 33.2     | 37       | 31.2     | 34.6     |
> > > | Source (Stat. (LN\&AF))                         | 42.1     | 29.4     | 19.5     | 27.6     | 31.2     | 29.4     | 29.9     |
> > > | + Tent (Dyn. LN)                                | 8.1      | 4        | 2.3      | 4.7      | 7.8      | 6.1      | 5.5      |
> > > | + EATA (Dyn. LN)                                | 46.7     | 30.5     | 28       | 31.4     | 35.4     | **33.8** | 34.3     |
> > > | + SAR (Dyn. LN)                                 | 43.1     | 17.3     | 8.3      | 29       | 31.6     | 30.5     | 26.6     |
> > > | + RFRA (SAF)                                    | **47.3** | **32.7** | **29.9** | **33.2** | **38.3** | 33.7     | **35.8** |
> > >
> > > > *Q1.2: Moreover, it would be beneficial to evaluate how the method performs in the context of test-time adaptation (TTA) **under unbiased reliability conditions**. Specifically, investigating the method's effectiveness in both (i.d.d. and non-i.d.d. scenarios would render it more practical and versatile.*
> > >
> > > **A1.2:** Thanks for your comment. We understand your concern and conduct additional experiments in the context of test-time adaptation (TTA) under unbiased reliability conditions. Specifically, we directly adapt RFRA and other baseline methods into the test sets of the VGGSound and Kinetics datasets without adding any corruptions. Results are summarized in Table 9 within the revised manuscript. For your convenience, we attach the corresponding results in the following table.
> > >
> > > | Method     | Source (Stat. (LN\&AF)) | Tent | EATA | SAR  | RFRA |
> > > | :--------- | :---------------------: | :--: | :--: | :--: | :--: |
> > > | VGGSound   |          63.3           | 62.6 | 63.1 | 63.1 | 63.5 |
> > > | Kinetics50 |          82.3           | 82.1 | 82.3 | 82.3 | 82.2 |
> > >
> > > The results indicate that the robustness of RFRA doesn't compromise the performance of the original target domain. In essence, RFRA remains effective even in scenarios where the presence of distribution shifts is uncertain.
> > >
> > > > *Q2.1: The paper contains extensive analysis and experiments, but some settings require further clarification. For instance, it's not entirely clear **what "Attention A-V" means** in Figure 4 and **how the results demonstrate the method's robustness**.*
> > >
> > > **A2.1: ** **What "Attention A-V" means.** "Attention A-V" denotes the cross-attention from the audio modality to the video modality. Specifically, in Figure 4, the attention values between modalities, denoted as "Attention X-Y" (X, Y $\in$ {A, V}), are derived using Equation 4 from the manuscript:
> > >
> > > $$\mathbf{A}=\operatorname{Softmax}\left(\frac{\mathbf{Q} \mathbf{K}^T}{\sqrt{d}}\right)$$.
> > >
> > > For instance, "Attention X-X" represents the self-attention, computed by setting both the query ($\mathbf{Q}$) and key ($\mathbf{K}$) as the tokens of modality X. Meanwhile, "Attention X-Y" (X, Y $\in$ {A, V}) corresponds to the configuration where the query and key tokens correspond to modality X and modality Y, respectively.
> > >
> > > **How the results demonstrate the robustness.** Larger attention values indicate a greater focus on a particular modality during cross-modal fusion. In Figure 4, "Tent" displays slightly better robust fusion effects compared to "AF," potentially due to the narrowed domain gap achieved by repurposing the LN. In contrast, our method demonstrates significant improvements in reliability estimation (attention values) for both video and audio bias situations across varying severities. This highlights the necessity of the self-adaptive attention-based fusion paradigm for multi-modal TTA, as it enhances the model's capability to adapt to reliability bias scenarios.

---

> ### Author Response · Authors · 2023-11-19
> **The Response to Reviewer 9FmQ (Part 4)**
>
> > *Q2.2: In the analysis of Figure 5, claims are made about the importance of maintenance between audio and video modalities, but **the contrast with clean results is not evident**. Additional clarification or supplementary results are needed to support these claims.*
>
> Thanks for your comment. We have supplemented more visualization results on different baselines for comparisons. Specifically, Fig. 11 in the Appendix of the manuscript shows the attention matrixes of the source model with vanilla attention-based fusion (AF), and the model adapted by Tent with dynamic LN.
>
> > *Q3: There are a few **typos and vague statements** in the paper, such as "trafﬁc noise in the audio modality" in the caption of Figure 1, "information bias" on Page 2, and inconsistent notations like "(Stat. LN) & AF" or "Stat. (LN & AF)" in Tables 1-3. These should be corrected for clarity and consistency.*
>
> We appreciate your feedback on the typos. In the updated version, we have carefully revised the typos.

---

> > ### Comment · Reviewer_9FmQ · 2023-11-22
> >
> > After reading the comments from other reviewer and the response from the authors, I would like to promote my score to 'accept'. This paper deserves to be presented on ICLR 2024.

---

> > > ### Author Response · Authors · 2023-11-22
> > > **Thank you for increasing your score!**
> > >
> > > We sincerely appreciate your positive recognition and assessment of our work!

---

> ### Author Response · Authors · 2023-11-22
> **Inquiry about further concerns/questions**
>
> Dear reviewer 9FmQ,
>
> As the author/reviewer discussion will draw to a close soon, we would like to know if our response has addressed your concerns and questions. If you have any further concerns or suggestions for the paper or our rebuttal, please let us know. We would be happy to engage in further discussion and manuscript improvement.
>
> Thank you again for the time and effort you dedicated to reviewing this work.

---

### Official Review · Reviewer_r3k4 · 2023-10-29

**Soundness:** 3 good
**Presentation:** 3 good
**Contribution:** 3 good
**Rating:** 8
**Confidence:** 4

**Summary:**

This paper delves into test-time adaption under the multi-modal setting and reveals an interesting and practical challenge, namely, reliability bias. In the wild, it is common that some modalities would suffer from distribution shifts compared to their counterparts in the source domain. As a result, the task-specific information across the modalities would be more inconsistent, thus contributing to the reliability differences for different modalities. Extensive empirical studies have been conducted to investigate the impact of the reliability bias using different cross-modal fusion strategies. To achieve robust multi-modal TTA against reliability bias,  the authors propose a novel method, dubbed reliable fusion and robust adaption (RFRA). Different from the existing TTA methods that mainly repurpose the normalization layers to achieve adaption, RFRA modulates the attention module to achieve reliable cross-modal fusion during test time. Besides, RFRA adopts a new objective function with desirable mathematical properties to combat with noise during adaption. To highlight the necessity of developing reliability-bias robust multi-modal TTA, the authors construct two new benchmarks with different settings of reliability bias based on the Kinetics and VGGSound datasets. Finally, the authors validate the effectiveness of the proposed method and give a deep analysis of the reliability bias challenge.

**Strengths:**

1. Revealing a new problem. This paper studies a new and practical challenge in the context of multi-modal test-time adaptation, namely, reliability bias. In the wild, it is evitably to introduce distribution shifts in some modalities.  As a result, the task-specific information across the modalities would be more inconsistent, thus contributing to the reliability differences for different modalities. The authors design extensive experiments to investigate the influence of reliability bias under different multi-modal fusion manners and the results validate the necessity of handling reliability bias for multi-modal TTA. I think the revealed challenge would bring some insights to the TTA community.
2. Constructing meaningful benchmarks. To highlight the necessity of developing reliability-bias robust multi-modal TTA, the authors construct two new benchmarks with different settings of reliability bias based on the Kinetics and VGGSound datasets. Concretely, the benchmarks consist of both video and audio modalities and each modality is with corruptions of different levels so that the reliability bias is simulated. On the one hand, the corruption on video modality follows ImageNet-C, which ensures comparison fairness. On the other hand, the incorporation of audio corruption types extends the utility of this research to the audio domain with TTA. Besides,  the benchmarks encompass diverse multi-modal tasks, including action recognition and event classification, thus providing a comprehensive evaluation and supplementing existing multi-modal test-time adaptation tasks.
3. Innovative paradigm for TTA. This paper proposes a new paradigm for TTA, namely, repurposing the attention layers during test time. Intuitively, the parameter updating of normalization layers like most existing TTA methods can only handle the distribution shifts between domains. In contrast, the parameter modulation of attention layers would help learn the importance difference between modalities, resulting in the reliable fusion for multi-modal TTA. The authors perform extensive experiments to show the superiority of the proposed new paradigm against reliability bias.

**Weaknesses:**

1. While the paper is well-written, it lacks essential details about CAV-MAE, such as the number of attention layers used for fusion. As many evaluations rely on the CAV-MAE framework, providing this information is crucial for the readers and reviewers to fully grasp the methodology. Additionally, visualizing the corruption types on both video and audio modalities would enhance the paper's clarity and help readers better understand the benchmarks.
2. The paper introduces the modulation of Q, K, and V parameters to address reliability bias. However, it's not clear why the authors chose to update all parameters of Q, K, and V simultaneously. Explaining this choice and discussing the possibility of updating only the MLP parameters of Q and K would provide valuable insights. Furthermore, while the novel attention layer repurposing is effective for reliability bias, the paper should address whether this approach comes at the cost of efficiency for test-time adaptation.
3. The paper introduces a crucial hyper-parameter, the confidence threshold ($\gamma$) in Equation 6. While this threshold is fixed in all experiments, it's essential to include ablation studies to explore the sensitivity of the proposed method to variations in this hyper-parameter. A more comprehensive analysis would provide a deeper understanding of the method's robustness.

Minor: Some of the figures, such as Figure 1c and Figure 3b, suffer from low image clarity. Improving the quality of these figures would enhance the paper's visual presentation and make the findings more accessible to readers.

Overall evaluation, I think this paper is above the bar of ICLR, regarding motivation and novelty.


----------------------------------

Upon reviewing the response, I note that my concerns have been effectively addressed. Considering the consensus of other reviewers, I wholeheartedly recommend this work with a high level of approval.

**Questions:**

My primary concerns revolve around the lack of comprehensive experiment details and the design of the modulation strategies, as highlighted in the weaknesses.

---

> ### Author Response · Authors · 2023-11-19
> **The Response to Reviewer r3k4 (Part 1)**
>
> Thanks for your valuable reviews. We would like to address your concerns one by one in the following.
>
> > *Q1.1: While the paper is well-written, it **lacks essential details about CAV-MAE**, such as the number of attention layers used for fusion. As many evaluations rely on the CAV-MAE framework, providing this information is crucial for the readers and reviewers to fully grasp the methodology.*
>
> **A1.1**: We apologize for the missing details on the backbone that confuses the reviewer. In the revised manuscript, we have supplemented the details of CAV-MAE ([A]) and more implementation details of our RFRA in Section C. For your convenience, we attach the added statement as follows.
>
> In the implementation, we use the CAV-MAE model as the backbone. CAV-MAE adopts an encoder-decoder-like architecture that is pre-trained on large-scale video data with both the contrastive learning and mask image modeling paradigms. The CAV-MAE encoder consists of 11 Transformer layers dedicated to each modality for the modality-speciﬁc feature extraction, alongside one Transformer layer for cross-modal fusion. The input to the CAV-MAE encoder involves 10-second video clips containing both video and corresponding audio data. For the video stream, CAVMAE samples 10 frames within each video clip and randomly selects one frame feeding into the visual Transformer encoder. For the audio stream, each 10-second audio waveform is converted into one spectrogram and then inputted to the audio Transformer encoder.
>
> During the ﬁne-tuning phase, we maintain the visual and audio encoders of the pre-trained model and add one randomly initialized classiﬁcation head to them. The ﬁne-tuned model is regarded as the source model and denoted as “Source (Stat. (LN & AF))”. To investigate the robustness of different fusion manners, we design another variant of the source model that utilizes 12 Transformer layers for feature extraction and performs late fusion between the classiﬁcation logit of each modality. The corresponding model variant is denoted as “Source ((Stat. LN) & LF)”. During the test-time adaption phase, unless otherwise speciﬁed, all baselines update the parameters of all normalization layers rooted in the source model, i.e., referred to as “Dyn. LN”. In contrast, as depicted in Fig. 2, our default approach in the RFRA framework involves updating only the parameters of the last Transformer layer to ensure reliable fusion, denoted as “SAF”.
>
> > *Q1.2: Additionally, **visualizing the corruption types on both video and audio modalities** would enhance the paper's clarity and help readers better understand the benchmarks.*
>
> **A1.2**: Thanks for your valuable suggestion. In this work, to comprehensively evaluate modality bias, we introduce different distribution shifts on the video and audio modalities for the test sets of VGGSound ([B]) and Kinetics ([C]) datasets. For the video corruptions, we follow [D] to apply 15 kinds of corruptions into the video, and each corruption has 5 kinds of severity levels for extensive validations. Specifically, the corruptions on video modality include ''Gaussian Noise", ''Shot Noise", ''Impulse Noise", ''Defocus Blur", ''Glass Blur", ''Motion Blur", ''Zoom Blur", ''Snow", ''Frost", ''Fog", ''Brightness", ''Elastic", ''Pixelate", ''Contrast", and ''JPEG". Similar to the video modality, we add 6 kinds of common audio noise (https://freesound.org) with 5 kinds of severity levels captured in the wild. Specifically, the corruptions on audio modality include ''Gaussian Noise", ''Paris Traffic Noise", ''Crowd Noise", ''Rainy Noise", ''Thunder Noise" and ''Windy Noise".
>
> In response to your valuable suggestion, we've incorporated visualizations of corruption types in both video and audio modalities within the updated manuscript. Fig. 6 showcases the visualization results of various visual corruption types observed within the constructed Kinetics-C benchmark. Additionally, Fig. 7 presents the Mel spectrogram visualizations, highlighting the raw audio and the corresponding audio corruption types. For a comprehensive understanding, we encourage you to refer to the updated manuscript for an in-depth exploration of these visualizations.

---

> > ### Author Response · Authors · 2023-11-19
> > **The Response to Reviewer r3k4 (Part 2)**
> >
> > > *Q2.1: The paper introduces the modulation of Q, K, and V parameters to address reliability bias. However, it's not clear why the authors chose to update all parameters of Q, K, and V simultaneously. **Explaining this choice and discussing the possibility of updating only the MLP parameters of Q and K** would provide valuable insights.*
> >
> > Thanks for your constructive comment. In our default approach, we update $W_{{\Theta}^h}$ and $B_{{\Theta}^h}$ $(h\in {q,k,v})$ within the last Transformer layer of the source model to ensure reliable fusion.
> >
> > In response to your valuable comment, we conduct additional experiments to explore the impact of different repurposing schemes. To this end, we design three variants: one that updates only the query and key projection layers, another that updates only the value projection layers, and a third that updates the final classification head. Results are summarized in Table 8 within the revised manuscript. For your convenience, we attach the corresponding results in the following table.
> >
> > | Corruption    | Source |  QK  |  V   | MLP  | QKV (ours) |
> > | :------------ | :----: | :--: | :--: | :--: | :--------: |
> > | Video-Fog     |  46.7  | 51.7 | 53.6 | 49.1 |    57.4    |
> > | Audio-Traffic |  65.5  | 68.8 | 67.2 | 66.7 |    69.0    |
> >
> > Results in the following table illustrate that the default setting, updating the query, key, and value projection layers simultaneously, exhibits significant performance superiority. Modulating the classification head demonstrates minimal effectiveness (e.g., from $46.7$ to $49.1$). Conversely, the attention modulation scheme achieves adaptive fusion between discrepant modalities, mitigating the multi-modal reliability bias problem (e.g., from $46.7$ to $51.7$). Moreover, modulation on the query, key, and value projection layers introduces additional parameters for reliable fusion, resulting in further improvements in robustness (e.g., from $46.7$ to $57.4$).
> >
> > > *Q2.2: Furthermore, while the novel attention layer repurposing is effective for reliability bias, the paper should address **whether this approach comes** **at the cost of efficiency for test-time adaptation**.*
> >
> > Thanks for your valuable suggestion. Different from most TTA methods that update the parameters of normalization layers, our RFRA repurposes the last Transformer layer of the CAV-MAE model as elaborated in Section 2 of the manuscript.
> >
> > In response to your insightful suggestion, we conduct additional experiments comparing the efficiency of the two paradigms. To this end, we choose the attention-fusion-based CAV-MAE model as the source model (\textit{i.e.}, source (Stat. (LN \& AF))), and conduct experiments on the VGGSound-C benchmark. We measure both the size of learnable parameters and the GPU time during the test-time adaptation phase. Results are summarized in Table 7 within the revised manuscript. For your convenience, we attach the corresponding results in the following table.
> >
> > | Method         | \#params (M) | GPU time (14,046 pairs) |
> > | :------------- | :----------: | :---------------------: |
> > | Tent (Dyn. LN) |     0.2      |      209.5 seconds      |
> > | EATA (Dyn. LN) |     0.2      |      207.6 seconds      |
> > | SAR (Dyn. LN)  |     0.2      |      286.1 seconds      |
> > | RFRA (SAF)     |     1.8      |      134.1 seconds      |
> >
> > The results underscore that RFRA achieves adaptation more efficiently, primarily due to its module repurposing approach. While the normalization layer updating scheme requires fewer parameters, it necessitates more time for propagation.
> >
> > > *Q3: The paper introduces a crucial hyper-parameter, the confidence threshold ($\gamma$) in Equation 6. While this threshold is fixed in all experiments, it's essential to include ablation studies to **explore the sensitivity of the proposed method to variations in this hyper-parameter**. A more comprehensive analysis would provide a deeper understanding of the method's robustness.*
> >
> > Thanks for your comment. In response to your concern, we investigate the influence of the only hyper-parameter (\textit{i.e.,} threshold $\gamma$ in Eq. 6) in our approach. To this end, we vary $\gamma$ in the range of $[0.1, 0.2, 0.3, e^{-1}, 0.4, 0.5]$ and perform corresponding experiments on the Kinetics50-C benchmark with fog and traffic corruptions. Results are depicted in Fig. 10 within the revised manuscript. For your convenience, we attach the corresponding numerical results in the following table.
> >
> > | Threshold ($\gamma$ in Eq. 6) | 0.1  | 0.2  | 0.3  | $e^{-1}$ | 0.4  | 0.5  |
> > | :---------------------------- | :--: | :--: | :--: | :------: | :--: | :--: |
> > | Video-Fog                     | 54.2 |  56  | 56.8 |   57.4   | 57.5 | 57.7 |
> > | Audio-Traffic                 | 69.2 | 69.1 | 69.1 |   69.0   | 69.3 | 69.0 |
> >
> > The results illustrates the stability of RFRA across varying threshold values of $\gamma$.

---

> > > ### Author Response · Authors · 2023-11-19
> > > **The Response to Reviewer r3k4 (Part 3)**
> > >
> > > > *Q4: Minor: Some of the figures, such as Figure 1c and Figure 3b, suffer from **low image clarity**. Improving the quality of these figures would enhance the paper's visual presentation and make the findings more accessible to readers.*
> > >
> > > We appreciate your feedback. In the revised manuscript, we have enhanced the quality of these figures to improve their visual presentation.
> > >
> > > **Reference:**
> > >
> > > [A] Yuan Gong, Andrew Rouditchenko, Alexander H Liu, David Harwath, Leonid Karlinsky, Hilde Kuehne, and James R Glass. Contrastive audio-visual masked autoencoder. In ICLR, 2023.
> > >
> > > [B] Honglie Chen, Weidi Xie, Andrea Vedaldi, and Andrew Zisserman. Vggsound: A large-scale audiovisual dataset. In ICASSP, 2020.
> > >
> > > [C] Will Kay, Joao Carreira, Karen Simonyan, Brian Zhang, Chloe Hillier, Sudheendra Vijayanarasimhan, Fabio Viola, Tim Green, Trevor Back, Paul Natsev, et al. The kinetics human action video dataset. arXiv:1705.06950, 2017.
> > >
> > > [D] Dan Hendrycks and Thomas Dietterich. Benchmarking neural network robustness to common corruptions and perturbations. In ICLR, 2019.

---

> ### Author Response · Authors · 2023-11-20
> **Thank you for increasing your score!**
>
> Thank you for upgrading your score! We appreciate the time and effort you dedicated to reviewing this work.

---

### Official Review · Reviewer_Hrwp · 2023-10-30

**Soundness:** 4 excellent
**Presentation:** 4 excellent
**Contribution:** 4 excellent
**Rating:** 8
**Confidence:** 5

**Summary:**

This work proposes a method for multi-modal test-time adaption (TTA) in the presence of cross-modal reliability bias. While there are numberer of works in test-time adaption, most of them focus on single-modality tasks and few works consider the practical multi-modal scenarios. In contrast, the authors investigate the characteristics of multi-modal TTA and reveal the task-specific cross-modal reliability bias setting where the information between modalities is unbalanced during test time derived from the distribution shifts across domains in some modalities. The authors conduct analysis experiments finding that the unreliable cross-modal fusion and noisy predictions hinder the robustness against cross-modal reliability bias. As a remedy, this paper proposes a novel method dubbed reliable fusion and robust adaption (RFRA). The idea of RFRA is straightforward. A self-adaptive attention is employed to achieve reliable cross-fusion during test time and a robust loss is adopted to prevent the prediction noise from dominating the adaption process. The authors experimentally validate their method on two new benchmarks against several baselines showing reasonable improvements.

**Strengths:**

1. This paper has a good motivation. The authors focus on multi-modal test-time adaption and reveal a NEW task-specific challenge (i.e., cross-modal reliability bias) for the first time. This paper first empirically proves that the existing test-time adaption methods cannot tackle the cross-modal reliability bias problem. Furthermore, the authors also investigate the effect of adopting the existing unbalanced multi-modal learning methods to handle the cross-modal information discrepancy.  The unbalanced multi-modal learning methods handle the unbalanced multi-modal data during training time with the help of labeled data, which resembles the test-time training paradigm in the domain adaption community. The results indicate that the multi-modal learning methods cannot take superiority for the challenge. In other words, the authors support the claim that designing an elaborated method for the cross-modal reliability bias during test time is essential.

2. The authors take an in-depth study on the cross-modal reliability bias challenge and find that robust cross-modal fusion and noise resistance are essential to achieve multi-modal TTA against reliability bias. To this end, the authors design a novel method dubbed reliable fusion and robust adaption (RFRA). On the one hand, RFRA achieves reliable fusion through the self-adaptive fusion module. On the other hand, RFRA employs an elaborately designed objective function to achieve noise-robust adaption. Figs. 1 and 2 clearly depict the motivation and key idea of the paper. Overall evaluation, this paper is with strong motivation, a technical sound approach, extensive experiments, and good writing.

**Weaknesses:**

1. The authors have conducted extensive evaluations on two newly constructed benchmarks regarding the most challenged setting (severity 5 in this paper), and the results indeed verify the effectiveness of handling the cross-modal reliability bias challenge. Even so, I think some challenging TTA settings that are orthogonal to the cross-modal reliability bias might help to improve the practicalness and impact of this work. First, in the practical multi-modal scenarios, the severity of distribution shifts might dynamically vary. It would make the work more practical if the authors could additionally investigate the robustness of the proposed method under the setting MIXED SEVERITIES. Furthermore, it is also common that the corruption types continually vary in the wild, resulting in the demand for continual test-time adaption (CTTA). The proposed approach would be more solid and universal if the proposed method could work in the MIXED DISTRIBUTION SHIFTS setting (i.e., continual TTA setting).

2. Some details are missing. It is not clear how many layers of the attention module are used and repurposed during test time adaption in the proposed approach. And how many parameters do these self-attention layers account for? It is encouraged to supplement comparisons with the TTA baselines regarding the number of modulated parameters and the adaption time, which would make the comparisons more comprehensive.

3. The results of the default setting can be added to the ablation tables for clear clarification. In the current form, the readers need to compare the main table (1,2,3) and the ablation tables for contrast.

4. I carefully read the paper and found two potential mathematical typos. $\theta_{s}^{m}$ in Line 5, page 4. $\partial$ in Eq. 9.

**Questions:**

My questions are mainly in the efficiency comparisons between the proposed SAF module and the TTA baselines, and some clarification on statements. Moreover, I wonder about the effect of the proposed method on the settings of MIXED SEVERITIES and MIXED DISTRIBUTION SHIFTS. Certainly, this is optional during the rebuttal time because the settings are out of the scope of the paper, but I think the results would strengthen this work.

---

> ### Author Response · Authors · 2023-11-19
> **The Response to Reviewer Hrwp (Part 1)**
>
> Thanks for your constructive reviews and suggestions. In the following, we will answer your questions one by one.
>
> > *Q1.1: I think some challenging TTA settings that are orthogonal to the cross-modal reliability bias might help to improve the practicalness and impact of this work. First, in the practical multi-modal scenarios, the severity of distribution shifts might dynamically vary. It would make the work more practical if the authors could **additionally investigate the robustness of the proposed method under the setting MIXED SEVERITIES**.*
>
> **A1.1**: In this paper, we reveal the multi-modal reliability challenge for the TTA community and propose a reliable fusion and robust adaption (RFRA) approach to tackle the issue. Following the widely-used TTA evaluation protocol ([A, B, C, D]), we introduce different corruption types to either the video or audio modality. In the main paper, we reported the performance under different corruption types with a severity level of 5.
>
> In response to your insightful suggestion, we conduct additional experiments on the Kinetics50-C benchmark, investigating the robustness of the proposed method under the setting of MIXED SEVERITY. To this end, we create test pairs for each corruption type by blending severity levels 1 through 5, resulting in 5N test pairs, where N represents the original size of the test data. After that, we shuffle the obtained test pairs and randomly choose N pairs for each corruption type. To verify the effectiveness of RFRA under the MIXED SEVERITY setting, we choose the typical TTA method Tent ([A]) and the SOTA TTA method SAR ([E]) as baselines for comparisons. The results are depicted in Fig. 8 within the revised manuscript. For your convenience, we attach the corresponding numerical result (regarding accuracy) in the following tables.
>
> | Kinetics50-C (Video Corruption, Mixed Severity) |  Gauss.  |   Shot   |  Impul.  |  Defoc.  |  Glass   |   Mot.   |   Zoom   |   Snow   | Frost    | Fog      | Brit.    | Digital  | Contr.   | Pix.     | JPEG     | AVG      |
> | :---------------------------------------------- | :------: | :------: | :------: | :------: | :------: | :------: | :------: | :------: | -------- | -------- | -------- | -------- | -------- | -------- | -------- | -------- |
> | Tent  ([A])                                     |   58.4   |   58.3   |   57.0   |   73.4   |   71.3   |   76.0   |   71.7   |   67.1   | 68.3     | 63.7     | **79.5** | 69.4     | 74.9     | 76.5     | 72.9     | 69.2     |
> | SAR ([E])                                       |   58.6   |   58.2   |   56.8   |   73.1   |   70.1   |   75.6   |   71.1   |   66.4   | 67.1     | 63.8     | 79.4     | 69.2     | 74.5     | 76.5     | 72.3     | 68.8     |
> | RFRA (Ours)                                     | **59.4** | **59.3** | **57.9** | **73.5** | **72.1** | **76.2** | **72.6** | **68.2** | **70.3** | **68.7** | 79.4     | **69.7** | **75.1** | **76.7** | **73.7** | **70.2** |
>
> | Kinetics50-C (Audio Corruption, Mixed Severity) |  Gauss.  |  Traff.  |  Crowd.  |   Rain   |  Thund.  |   Wind   |   AVG    |
> | :---------------------------------------------- | :------: | :------: | :------: | :------: | :------: | :------: | :------: |
> | Tent  ([A])                                     |   76.1   |   68.9   |   70.3   |   70.4   |   68.0   | **72.5** |   71.0   |
> | SAR ([E])                                       |   76.1   |   66.3   |   69.1   |   69.1   |   68.5   |   72.2   |   70.2   |
> | RFRA (Ours)                                     | **76.5** | **71.2** | **71.2** | **72.2** | **72.6** | **72.5** | **72.7** |
>
> The results indicate the effectiveness of RFRA in addressing cross-modal reliability bias across various corruption types exhibiting mixed severity levels.

---

> ### Author Response · Authors · 2023-11-19
> **The Response to Reviewer Hrwp (Part 2)**
>
> > Q1.2: Furthermore, it is also common that the corruption types continually vary in the wild, resulting in the demand for continual test-time adaption (CTTA). The proposed approach would be more solid and universal if the proposed method could work **in the MIXED DISTRIBUTION SHIFTS setting (i.e., continual TTA setting)**.
>
> **A1.2**: Following the setting of continual TTA ([F, G]), we conduct additional experiments on the Kinetics50-C benchmark under the MIXED DISTRIBUTION SHIFTS setting as the reviewer suggested. In this setting, both baseline methods (Tent and SAR) alongside our RFRA continually adapt to evolving corruption types, and the averaged performance across all corruption types is reported. To ensure comprehensive evaluations, we vary the severity levels from 1 to 5. The results are summarized in Fig. 9 within the updated manuscript.
> For your convenience, we attach the corresponding numerical result (regarding accuracy) in the following tables.
>
> | Kinetics50-C (Video Corruption, Mixed Distribution Shifts) | Severity 1 | Severity 2 | Severity 3 | Severity 4 | Severity 5 | AVG      |
> | ---------------------------------------------------------- | :--------: | :--------: | :--------: | :--------: | :--------: | -------- |
> | Tent  ([A])                                                |     76     |    72.1    |    67.1    |    39.5    |    38.3    | 58.6     |
> | SAR ([E])                                                  |    76.1    |    71.8    |  **69.0**  |    64.3    |    59.6    | 68.2     |
> | RFRA (Ours)                                                |  **76.8**  |  **72.5**  |  **69.0**  |  **65.0**  |  **61.7**  | **69.0** |
>
> | Kinetics50-C (Audio Corruption, Mixed Distribution Shifts) | Severity 1 | Severity 2 | Severity 3 | Severity 4 | Severity 5 |   AVG    |
> | -------------------------------------------------- | :--------: | :--------: | :--------: | :--------: | :--------: | :------: |
> | Tent  ([A])                                        |    71.2    |    70.8    |    70.0    |    69.4    |    68.5    |   70.0   |
> | SAR ([E])                                          |    71.0    |    70.6    |    70.0    |    69.8    |    69.3    |   70.1   |
> | RFRA (Ours)                                        |  **72.2**  |  **71.6**  |  **71.0**  |  **70.4**  |  **69.7**  | **71.0** |
>
> Notably, the performance of the vanilla TTA method (Tent) significantly degrades, especially under high-level mixed distribution shifts (e.g., 38.3 at Severity 5). In contrast, our RFRA showcases relative robustness against these shifts across different severity levels. It's worth highlighting that while our RFRA was primarily designed to address the challenge of multi-modal reliability bias, it demonstrates consistent performance superiority compared to SAR, an approach explicitly tailored for handling mixed distribution shifts in the continual TTA setting. This observation underscores the adaptability and resilience of RFRA beyond its initial design scope.
>
> >  *Q2.1: **Some details are missing**. It is not clear how many layers of the attention module are used and repurposed during test time adaption in the proposed approach. And how many parameters do these self-attention layers account for?*
>
> **A2.1**: We apologize for the initial oversight in providing comprehensive details. In the revised manuscript, we have supplemented the details of CAV-MAE ([H]) and implementation details of our RFRA. For your convenience, we attach the added statement as follows.
>
> In the implementation, we use the CAV-MAE model as the backbone. CAV-MAE adopts an encoder-decoder-like architecture that is pre-trained on large-scale video data with both the contrastive learning and mask image modeling paradigms. The CAV-MAE encoder consists of 11 Transformer layers dedicated to each modality for the modality-speciﬁc feature extraction, alongside one Transformer layer for cross-modal fusion. The input to the CAV-MAE encoder involves 10-second video clips containing both video and corresponding audio data. For the video stream, CAVMAE samples 10 frames within each video clip and randomly selects one frame feeding into the visual Transformer encoder. For the audio stream, each 10-second audio waveform is converted into one spectrogram and then inputted to the audio Transformer encoder.

---

> > ### Author Response · Authors · 2023-11-19
> > **The Response to Reviewer Hrwp (Part 3)**
> >
> > During the ﬁne-tuning phase, we maintain the visual and audio encoders of the pre-trained model and add one randomly initialized classiﬁcation head to them. The ﬁne-tuned model is regarded as the source model and denoted as “Source (Stat. (LN & AF))”. To investigate the robustness of different fusion manners, we design another variant of the source model that utilizes 12 Transformer layers for feature extraction and performs late fusion between the classiﬁcation logit of each modality. The corresponding model variant is denoted as “Source ((Stat. LN) & LF)”. During the test-time adaption phase, unless otherwise speciﬁed, all baselines update the parameters of all normalization layers rooted in the source model, i.e., referred to as “Dyn. LN”. In contrast, as depicted in Fig. 2, our default approach in the RFRA framework involves updating only the parameters of the last Transformer layer to ensure reliable fusion, denoted as “SAF”.
> >
> > > *Q2.2: It is encouraged to supplement **comparisons with the TTA baselines regarding the number of modulated parameters and the adaption time**, which would make the comparisons more comprehensive.*
> >
> > **A2.2:** Thanks for your valuable suggestion. Different from most TTA methods that update the parameters of normalization layers, our RFRA repurposes the last Transformer layer of the CAV-MAE model as elaborated in Section 2 of the manuscript. In response to your insightful suggestion, we conduct additional experiments comparing the efficiency of the two paradigms. To this end, we choose the attention-fusion-based CAV-MAE model as the source model (\textit{i.e.}, source (Stat. (LN \& AF))), and conduct experiments on the VGGSound-C benchmark. We measure both the size of learnable parameters and the GPU time during the test-time adaptation phase. Results in the following table highlight that our RFRA accomplishes adaptation in less time. The efficiency of RFRA can be attributed to its module repurposing approach. Although the normalization layer updating scheme occupies fewer parameters, it demands more time for propagation.
> >
> > | Method         | \#params (M) | GPU time (14,046 pairs) |
> > | :------------- | :----------: | :---------------------: |
> > | Tent (Dyn. LN) |     0.2      |      209.5 seconds      |
> > | EATA (Dyn. LN) |     0.2      |      207.6 seconds      |
> > | SAR (Dyn. LN)  |     0.2      |      286.1 seconds      |
> > | RFRA (SAF)     |     1.8      |      134.1 seconds      |
> >
> >
> >
> > > *Q3: The results of **the default setting can be added** to the ablation tables for clear clariﬁcation. In the current form, the readers need to compare the main table (1,2,3) and the ablation tables for contrast.*
> >
> > **A3**: We apologize for the confusion arising from the initial presentation. To address this concern, we've made necessary adjustments in the revised manuscript by incorporating the default setting's results, distinctly highlighted in pink, within the ablation tables (Tables 4, 5, and 6).
> >
> > > *Q4: I carefully read the paper and found two potential **mathematical typos**. $\theta^{m}_{s}$ in Line 5, page 4. $\partial$ in Eq. 9.*
> >
> > **A4**:  Thanks for your careful reading. We apologize for the typos and have revised them in the updated manuscript.
> >
> > **Reference:**
> >
> > [A] Dequan Wang, Evan Shelhamer, Shaoteng Liu, Bruno Olshausen, and Trevor Darrell. Tent: Fully test-time adaptation by entropy minimization. In ICLR, 2021.
> >
> > [B] Yu Sun, Xiaolong Wang, Zhuang Liu, John Miller, Alexei Efros, and Moritz Hardt. Test-time training with self-supervision for generalization under distribution shifts. In ICML, 2020.
> >
> > [C] Shuaicheng Niu, Jiaxiang Wu, Yifan Zhang, Yaofo Chen, Shijian Zheng, Peilin Zhao, and Mingkui Tan. Efﬁcient test-time model adaptation without forgetting. In ICML, 2022.
> >
> > [D] Inkyu Shin, Yi-Hsuan Tsai, Bingbing Zhuang, Samuel Schulter, Buyu Liu, Sparsh Garg, In So Kweon, and Kuk-Jin Yoon. Mm-tta: multi-modal test-time adaptation for 3d semantic segmentation. In CVPR, 2022.
> >
> > [E] Shuaicheng Niu, Jiaxiang Wu, Yifan Zhang, Zhiquan Wen, Yaofo Chen, Peilin Zhao, and Mingkui Tan. Towards stable test-time adaptation in dynamic wild world. In ICLR, 2023.
> >
> > [F] Yulu Gan, Yan Bai, Yihang Lou, Xianzheng Ma, Renrui Zhang, Nian Shi, and Lin Luo. Decorate the newcomers: Visual domain prompt for continual test time adaptation. In AAAI, 2023.
> >
> > [G] Qin Wang, Olga Fink, Luc Van Gool, and Dengxin Dai. Continual test-time domain adaptation. In CVPR, 2022.
> >
> > [H] Yuan Gong, Andrew Rouditchenko, Alexander H Liu, David Harwath, Leonid Karlinsky, Hilde Kuehne, and James R Glass. Contrastive audio-visual masked autoencoder. In ICLR, 2023.

---

### Meta-Review · Area_Chair_6XCe · 2023-12-06

**Metareview:**

This paper presents a new approach for addressing reliability bias in multi-modal test-time adaptation. The research problem is well motivated, and the proposed method is technically sound. The paper also introduces two new benchmark datasets, and experimental results are extensive and convincing. Reviewers raised some concerns about technical details, experimental results, ablation studies, and paper writing, which have been adequately addressed in the authors' responses. In addition, some descriptions in the paper are unclear, such as: "Comparisons between different fusion manners", "The blocks of the top left and bottom right denote", and "Pink denote the default setting." The authors are encouraged to proofread the paper carefully.

**Justification For Why Not Higher Score:**

Reviewers raised some comments on technical details and experimental details, which should be carefully addressed in the final version.

**Justification For Why Not Lower Score:**

This paper studies a new problem, i.e., reliability bias for multi-modal test-time adaptation. The paper is well written.

---

### Decision · Program_Chairs · 2024-01-16

Accept (poster)